# Mixed Mulberry Fruit and Mulberry Leaf Fermented Alcoholic Beverages: Assessment of Chemical Composition, Antioxidant Capacity In Vitro and Sensory Evaluation

**DOI:** 10.3390/foods11193125

**Published:** 2022-10-08

**Authors:** Tengqi Gao, Jinling Chen, Feng Xu, Yilin Wang, Pengpeng Zhao, Yunfei Ding, Yongbin Han, Jie Yang, Yang Tao

**Affiliations:** 1Co-Innovation Center of Jiangsu Marine Bio-industry Technology, Jiangsu Ocean University, Lianyungang 222005, China; 2Jiangsu Key Laboratory of Marine Bioresources and Environment/Jiangsu Key Laboratory of Marine Biotechnology, Jiangsu Ocean University, Lianyungang 222005, China; 3College of Food Science and Technology, Nanjing Agricultural University, Nanjing 210095, China

**Keywords:** mulberry leaves, *Saccharomyces cerevisiae*, mixed fermentation, functional substance, Gas chromatography-mass spectrometer

## Abstract

The fruit and leaves of mulberry (*Morus alba* L.) contain a variety of health-beneficial components. At present, although both alcoholic and non-alcoholic mulberry beverages are commercially available, studies on fermented alcoholic beverages mixed with both mulberry fruit and mulberry leaf are rare. In this study, different amounts (1, 2 and 3%, g/mL dry weight) of mulberry leaves were added during the alcoholic fermentation of mulberry juice. After 9 days of fermentation, the 1-deoxynojirimycin (DNJ) content increased from 61.12 ± 3.10 to 153.39 ± 3.98 μg/mL, and the quercetin content increased from 0.45 ± 0.01 to 20.14 ± 0.08 mg/L in the mulberry alcoholic beverages with the addition of mulberry leaves at 2%. Moreover, the ABTS^+^ scavenging capacity at the end of fermentation for the same sample was enhanced by 40.9%. In addition, the total sugar, total phenols, total anthocyanins, and γ-aminobutyric acid (GABA) contents of the fermented samples all decreased significantly at the end of fermentation. A total of 33 volatile compounds and 17 free amino acids were detected in the fermented alcoholic beverages with mulberry leaves added. The total free amino acid content increased with the increase in mulberry leaf addition. Principal component analysis showed that the addition of mulberry leaves during fermentation increased the contents of GABA, DNJ, total flavonols and protocatechuic acid in mulberry alcoholic beverages. All these studies revealed the dynamic changes in functional components in the alcoholic fermentation of mulberry juices with the addition of mulberry leaves. Overall, the addition of mulberry leaf powder at 2% was selected as the appropriate addition for producing mulberry alcoholic beverages with enhanced nutritional value.

## 1. Introduction

The leaves of the mulberry (*Morus alba* L.) are rich in biologically active ingredients, including phenolics, alkaloids and amino acids. These components have numerous health benefits, including antioxidant, antiobesity, antiatherosclerosis, anti-inflammatory and antidiabetic activities [1,2]. One of these components, 1-deoxynojirimycin (DNJ), is a natural alkaloid that is present in high concentrations in mulberry, particularly in mulberry leaves. DNJ is an α-glucosidase inhibitor and can lower blood sugar without intestinal side effects [3]. Mulberry leaves also contain high concentrations of γ-aminobutyric acid (GABA) which acts as a neuroinhibitory amino acid and sedative and lowers blood pressure [4]. Mulberry leaves have long been used in medicine to treat fever, protect the liver, brighten the eyes, improve joint function and diuresis and lower blood pressure [5]. This indicates that mulberry leaves have significant potential in a variety of applications. Various countries, including Japan, allow the sale of foods with mulberry leaves in markets, and other countries have allowed foods with mulberry. These foods, such as mulberry leaf biscuits, mulberry tea, mulberry leaf juice beverages and other products, are sold as functional foods specifically for lowering blood pressure. China has also developed a variety of mulberry leaf health foods and functional beverages, such as mulberry leaf health wine and mulberry leaf oral liquid [6].

Mulberry fruit is a berry with a water content of higher 80% when ripe. Mulberry fruit is rich in glucose, sucrose and fructose, with a total sugar content of 12−15%; it is also rich in flavonoids, anthocyanins, phenolic acids and other polyphenolic bioactive substances [7,8]. It has antioxidant, antifatigue, anti-inflammatory, neuroprotective and anticancer effects [9,10].

Our previous study found that the nutrient contents in mulberry wines were well preserved, which suggests that mulberry wine has a huge market potential [11]. In addition, due to mulberry leaves being rich in nutrients, mulberry leaf beverages and mulberry leaf health fermented beverages have appeared on the market recently, but there are no scientific reports of fermented beverages prepared by mixing mulberry fruits and mulberry leaves. In this study, based on the previous research on the characteristics of mulberry wine, the dynamic changes in functional components during the alcoholic fermentation mulberry fruit juice containing mulberry leaf powders were studied to clarify the effects of alcoholic fermentation on phenolics, DNJ and amino acids. This study provides a theoretical basis for the rational development and production of novel mulberry-based beverage with enhanced nutritional value.

## 2. Materials and Methods

### 2.1. Fresh Mulberry Fruits, Mulberry Leaves and Strain

Fresh mulberry fruits and mulberry leaves of the mulberry variety “*Humulus*” were kindly provided by Jiujiu Silk Company from Suqian, China. The mulberry fruits and leaves were harvested each year in May. After picking, the fruits and leaves were immediately transported to the laboratory and frozen at −20 °C. Mulberry leaves of similar lengths were selected, wiped, dried at 60 °C to constant weight (moisture content: 8.79 ± 0.23%) and pulverized (particle size: D50 = 86.78 ± 6.32 μm). There were three replicates of fermentation samples.

*Saccharomyces cerevisiae* ySR 127 (French yeast) was provided by Jiangsu Academy of Agricultural Sciences, Nanjing, China.

### 2.2. Chemicals and Reagents

All standards were purchased from Shanghai Yeyuan Biotechnology Co., Ltd., Shanghai, China and chromatographic-grade acetonitrile, methanol and trichloroacetic acid (TCA) were purchased from Sinopharm Chemical Reagent Co., Ltd., Shanghai, China. The Accela liquid chromatography system was purchased from Thermo Scientific (Waltham, MA, USA).

### 2.3. Preparation of Mixed Mulberry Fruit and Mulberry Leaf Fermented Alcoholic Beverages

The preparation of the mixed mulberry fruit and mulberry leaf fermented alcoholic beverages followed the methods of Duarte et al. [12] and Juan et al. [13], with certain modifications. Mulberry fruits of the same color and maturity were selected, homogenized and centrifuged at 10,000× *g* for 10 min to obtain the juice. The total soluble solid content was adjusted to 22° Brix with sucrose, the pH was adjusted to 3.5 with citric acid and the SO_2_ concentration was adjusted to 69 mg/L with K_2_S_2_O_5_.

Different amounts, namely, 1, 2 and 3% (*w*/*v*, g/mL dry weight) of the powdered mulberry leaves were added to the juice; mulberry beverages without added leaves were used as the control. *S. cerevisiae* ySR 127 were subcultured in YPD liquid medium at 28 °C and 200 r/min shaking for 48 h. The yeast was inoculated into the juice to a concentration of 4%, with an initial yeast concentration of approximately 6.0 log CFU/mL. The juice was then fermented at 25 °C in the dark for nine days. During the fermentation, the flask was shaken every day to ensure full mixing of the leaf powder and the juice. Samples of the beverages were collected for analysis at 0, 1, 3, 5, 7 and 9 days [11]. The initial value of the control sample was also a reference for all the samples (day 0). After centrifugation (10,000× *g*, 10 min), the supernatants were retained and stored at −20 °C. Samples were collected and analyzed in triplicate.

### 2.4. Chemical Analysis

#### 2.4.1. Determination of Basic Oenological and Chemical Parameters

The alcohol content, volatile acidity, total SO_2_ concentration, free SO_2_ concentration and pH were measured using the OIV official analytical methods in the samples on day 9 [14]. In addition, the total sugar content of the mulberry alcoholic beverages was determined using the sulfuric acid–anthrone method [15], and the results were expressed as mg/mL of glucose equivalents.

#### 2.4.2. Spectrophotometric Measurements of Phenolics, Color and Antioxidant Activity

The total phenolic content was determined by the Folin–Ciocalteu method [16]. One milliliter of the sample was added to 5 mL of water, after which 3 mL of 7.5% Na_2_CO_3_ was added. Finally, 1 mL of 0.1 M Folin phenolic reagent was added, and the solution was shaken well. The solution was allowed to stand for 1 h in the dark and the absorbance at 765 nm was then measured. The standard curve was prepared with gallic acid, and the results were expressed as mg GAE/g DW mulberry fermented alcoholic beverages.

Total anthocyanins content was determined by spectrophotometry method described by Ivanova et al. [17]. Ethanol, water and hydrochloric acid were mixed at a volume ratio of 69:30:1 and prepared into a solution. The sample was diluted with this solution and the absorbance value was measured at 540 nm. Total anthocyanins content was calculated according to the following formula:TA_540nm_ = A_540nm_ × 16.7 × d(1)
where A is the absorbance value and d is the dilution ratio of the sample. The results were calculated as mg mallow malvidin-3-*O*-glucoside/L mulberry fermented alcoholic beverages.

Monomeric anthocyanins, polymeric anthocyanins and total anthocyanins were measured according to the influence of SO_2_ and acetaldehyde on the chroma of different forms of anthocyanins [18]. After adding 20 μL 20% acetaldehyde to a 2 mL fermentation sample, the sample was left standing for 45 min, and then the absorbance value at 520 nm was measured and written as A^ace^; after adding 160 μL 5% SO_2_ to a 2 mL fermented sample, the absorbance value at 520 nm was measured and denoted as A^SO2^. A 2 mL sample was taken, and the absorbance value at 520 nm was measured and written as A^wine^. The absorbance value was measured with a 1 mm quartz cuvette and corrected with a 1 cm cuvette according to the following formulae:

Monomer anthocyanin = A^wine^ − A^SO2^

Polymeric anthocyanin = A^SO2^

The total anthocyanins = A^ace^

The absorbance value (AU) was used to represent the fermented sample results.

The tartaric esters and total flavonol contents were determined by spectrophotometry [18]. The fermented beverage sample (0.5 mL) was diluted with 10% ethanol to 5 mL, and 0.25 mL of the solution was mixed with 0.25 mL of 0.1% HCl dissolved in 95% ethanol. After the addition of 4.55 mL of 2% HCl, the solution was vortexed and allowed to stand for 15 min. Absorbances at 320 and 360 nm were then measured to determine the tartrate and total flavonol contents, respectively. The flavonol standard curve was prepared with quercetin dissolved in 95% ethanol. The tartaric ester standard curve was prepared with caffeic acid standards dissolved in 10% ethanol. The flavonol and tartrate results were calculated as mg RE/g DW mulberry fermented alcoholic beverages and mg caffeic acid/L mulberry fermented alcoholic beverages, respectively.

The contents of color were measured using our previous method [11].

The ABTS^+^ free-radical-scavenging activity was determined using the method of Liang et al. [19]. Equal volumes of 7 mM ABTS^+^ solution and 2.45 mM potassium persulfate were mixed and allowed to stand at room temperature in the dark for 12–16 h. The ABTS^+^ solution was then diluted with 0.2 M PBS (pH 7.4) until the absorbance value at 734 nm was 0.70 ± 0.02. For sample determination, 0.2 mL of the sample solution was mixed with 3.8 mL ABTS^+^ working solution and allowed to stand for 6 min at room temperature in the dark, after which the absorbance at 734 nm was measured. Trolox (TE) was used as the standard, and the results were expressed as μmol TE/L mulberry fermented alcoholic beverages.

Ferric antioxidant power (FRAP) was determined using the method of He et al. [20]. FRAP solution (10 mM FRAP in 40 mM HCl), 20 mM FeCl_3_ solution and 0.3 M acetic acid buffer (pH 3.6) were mixed at 1:1:10 (*v/v/v*) and incubated in a water bath at 37 °C for 1 h. Then, 0.1 mL of sample was added to 0.3 mL of water, and the diluted sample was added to 3 mL of the FRAP reagent and was mixed evenly. Absorbances were measured at 593 nm after a 30 min reaction in a 37 °C water bath. Fe_2_SO_4_ was used as the standard, and the results were expressed as μmol Fe^2+^/L mulberry fermented alcoholic beverages.

#### 2.4.3. Quantitative Determination of Phenolic Compounds by High Performance Liquid Chromatography (HPLC)

The anthocyanin compositions and contents were determined by HPLC (Agilent 1200; GMI, Saint Paul, MN, USA) [21] using an Agilent TC-C18 column (4.6 × 250 mm, 5 μm) with a mobile phase A of 0.5% TCA and a mobile phase B of acetonitrile. The column temperature was 30 °C, and the gradient was set at a flow rate of 0.5 mL/min as follows: 0–5 min, 10−12% B; 5–14 min, 12−13% B; 14–16 min, 13−14% B; 16–18 min, 14−16% B; 18–19 min, 16−18% B; 19–22 min, 18–22% B; 22–35 min, 22−30% B; the injection volume was 20 μL.

The extraction of phenolic acids was based on the method of Asadi et al. [22]. The pH of the sample was first adjusted to 2.0, and the suspension was saturated with NaCl, followed by three 30 min extractions with ethyl acetate (1:1, *v/v*). The organic phases in the upper layers after the three extractions were combined and dried in a rotary evaporator. The material remaining after evaporation was dissolved in methanol and filtered through a 0.45 μm filter membrane for HPLC analysis. The extraction of flavonol compounds was performed according to the phenolic acid extraction method; the difference was that the pH of the sample was first adjusted to 7.0.

For the extraction of flavonol compounds, the pH of the mulberry fermented alcoholic beverages was adjusted to 7.0, and the extraction followed the procedure used for the extraction of phenolic acids.

HPLC analysis of phenolic acids and flavonols was performed as previously described [23] using an Agilent Zorbax Eclipse SB-C18 (4.6 × 250 mm, 5 μm) column at a temperature of 25 °C. The mobile phase A was 1% acetic acid aqueous solution and the mobile phase B comprised 1% acetic acid methanol solution, with a flow rate of 0.6 mL/min. The gradient was as follows: 0–10 min, 10–26% B; 10–25 min, 26–40% B; 25–45 min, 40–65% B; 45–55 min, 65–95% B; 55–58 min, 95–10% B; 58–61 min, 10% B. The absorbance wavelength used for phenolic acids and dihydroquercetin was 280 nm, while flavonols were detected at 350 nm; the injection volume was 20 μL.

The standard curves of the mixed standard were determined using the external standard method, and the standard lines of the two monomeric anthocyanins were as follows: cyanidin-3-*O*-glucoside, y = 42,853x + 23, R^2^ = 0.9991; cyanidin-3-*O*-rutinoside, y = 26,647x + 36, R^2^ = 0.9997, where y is the peak area and x unit is mg/mL.

The standard curves of five monomeric phenolic acids were as follows: protocatechuic acid, y = 45,370x + 67, R^2^ = 0.9994; p-hydroxybenzoic acid, y = 43,489x + 48, R^2^ = 0.9996; caffeic acid, y = 104,479x + 89, R^2^ = 0.9991; 4-hydroxycinnamic acid, y = 157,189x + 54, R^2^ = 0.9997; resveratrol, y = 52,941x + 47, R^2^ = 0.9996, where y is the peak area and x unit is mg/mL.

The standard curves of five monomeric flavonoids were as follows: rutin, y = 79,968x + 24, R^2^ = 0.9998; myricetin, y = 90,421x + 43, R^2^ = 0.997; quercetin, y = 93,522x + 56, R^2^ = 0.998; kaempferol, y = 103,632x + 48, R^2^ = 0.9979; dihydroquercetin, y = 73,313x + 87, R^2^ = 0.9956, where y is the peak area and x unit is mg/mL.

#### 2.4.4. Determination of 1-Deoxynojirimycin (DNJ) Content

The DNJ content in fermented samples was determined by the method of Kim et al. [24]. One milliliter of fermented sample was accurately measured into a 10 mL centrifuge tube, followed by the addition of 2.5 mL of 0.05 mol/L hydrochloric acid solution. The solution was vortexed for 15 s and centrifuged at 12,000× *g* for 15 min. After collection of the precipitate, 2.5 mL of 0.05 mol/L hydrochloric acid was added and the extraction was repeated. The supernatant was collected twice and diluted to a constant volume of 6 mL with distilled water. The sample was derivatized with FMOC-Cl and analyzed using HPLC (Agilent 1200, GMI, Saint Paul, MN, USA) with a Waters XTERRA MS C18 column (150 × 4.6 mm, 5 μm). The column temperature was 25 °C and the detection wavelength was 254 nm. The mobile phase was acetonitrile and 0.1% acetic acid (55:45, *v/v*), the flow rate was 1.0 mL/min and the injection volume was 50 μL. DNJ was used as the standard, and the results were expressed as μg DNJ/mL fermented sample.

#### 2.4.5. Determination of γ-Aminobutyric Acid (GABA) Content

GABA in fermented samples was extracted according to the method of Bai et al. [25], and the samples were processed according to the method of Syu et al. [26]. Two milliliters of NaHCO_3_ solution were added to the fermented samples and centrifuged at 6000× *g* for 10 min. One milliliter of the supernatant was mixed with the same volume of amino acid derivative (4 mg/mL 4-dimethylaminoazobenzenesulfonyl chloride acetone solution) and allowed to react in a 67 °C water bath for 10 min. The reaction was rapidly terminated in an ice bath. The reaction solution was centrifuged at 6000× *g* for 20 min, and the supernatant was filtered through a 0.45 µm membrane for HPLC analysis. The chromatographic column was an Agilent SB-C18 column (4.6 × 250 mm, 5 μm), the mobile phase A was 0.045 mol/L acetic acid–sodium acetate buffer (pH 4.0), the mobile phase B was acetonitrile and the column temperature 30 °C. The detection wavelength was 425 nm and the injection volume was 20 µL. The gradient was set at a flow rate of 1 mL/min as follows: 0–10 min, 70% A; 10–12 min, 70−62% A; 12–20 min, 62% A; 20–22 min, 62−55% A; 22–30 min, 55% A; 30–36 min, 55−40% A; 36–40 min, 60% A; 40–41 min, 40−30% A; 41–50 min, 30% A. The results were expressed as μg GABA/mL fermented sample.

#### 2.4.6. Determination of Free Amino Acids

The free amino acids in the mulberry alcoholic beverages were determined according to the method of Aro et al. [27]. The samples were mixed with 4% trichloroacetic acid in a 1:1 (*v/v*) ratio and allowed to stand at 37 °C for 30 min. The samples were then centrifuged (15,000× *g*, 30 min) and passed through a 0.22 μm aqueous filter membrane. Finally, the supernatant was analyzed with a Hitachi L-8900 automatic amino acid analyzer (Hitachi Ltd., Tokyo, Japan). The analytical conditions were as follows: analytical column, 2622PH 4.6 mm I.D. × 60 mm; flow rate, 0.40 mL/min; column temperature, 57 °C; reaction temperature, 135 °C; detection wavelength, 570 nm; injection volume, 20 μL.

#### 2.4.7. Determination of Volatile Substances

Samples were collected at the end of fermentation and the volatile substances were analyzed by GC-MS (Agilent 7890B gas chromatography and Agilent 5977A mass spectrometer, Santa Clara, CA, USA), according to the method of Aznar and Arroyo [28]. The purge and capture extraction conditions were as follows: sample volume, 5 mL; dilution ratio of water to sample, 1:4 (*v*/*v*). The volatile substances were analyzed using the Atomax Teklink software control purging sample concentration system and then passed through N_2_ at a rate of 40 mL/min for 11 min at room temperature, with a desorption time of 5 min at 250 °C.

After extraction, the samples were analyzed by GC-MS. The gas chromatographic conditions were as follows: The column was an Agilent J & W DB-624 super-inert capillary column (30 m × 250 μm × 1.4 μm), carrying He gas at a velocity of 1 mL/min. The initial column temperature was kept at 35 °C for 2 min, after which it was raised to 120 °C at a rate of 5 °C/min and then to 220 °C at a rate of 10 °C/min. The column temperature was then kept at 220 °C for 2 min. The MS conditions included an ionization energy of 70 eV and an MS scanning range of 35–550 *m/z*.

### 2.5. Sensory Evaluation

Sensory evaluation was performed on samples taken on day 9 at the end of fermentation. Before the assessment, a round-table session was held to select sensory indicators for describing the quality of mulberry wine. The fermented alcoholic beverages were evaluated on three aspects: appearance (color and clarity), aroma (purity, fruity, alcohol, leaf, chocolate and intensity) and palate (body, alcohol, fruity, sweet, sour, tannin, yeast, mint, smoky and finish). The strength of each attribute was scored on a 0–10 scale. For each sample, 20 mL of fermented alcoholic beverage were poured into a 210 mL ISO tasting glass. Each sample was randomly coded with 3 Arabic numerals, the code also being written on the wine glass, and placed in random order. Sensory evaluation was carried out according to Tao et al. [29]. The sensory evaluation team consisted of ten members (five male and five female) from Shanghai Maotai Haima Enterprise Development Co., Ltd., a company specializing in the international wine trade. All the group members had attended training courses organized by the well-known Wines and Spirit Education Trust (WSET), and the assessment was performed in individual booths. These samples were assessed in duplicates.

### 2.6. Data Statistics and Analysis

All analyses were performed at least in duplicate and data were presented as means ± standard deviations. One-way analysis of variance (ANOVA) was performed with the Duncan method, and significant difference was expressed at the *p* < 0.05 level. Excel 2010, IBM SPSS statistical software (Version 26, IBM Company, Armonk, NY, USA) and Origin 2021 (OriginLab, Northampton, MA, USA) were used in these analyses.

## 3. Results and Discussion

### 3.1. Analysis of Basic Oenological and Chemical Parameters of the Mixed Mulberry Fruit and Mulberry Leaf Fermented Alcoholic Beverages

The basic oenological parameters of the mixed mulberry fruit and mulberry leaf fermented alcoholic beverages are summarized in Table 1. The alcohol contents of fruit wines are usually between 8 and 14%; all four samples were within this range, with no significant difference in the alcohol contents of the four samples [30]. However, excessive consumption can still cause harm to the body. In general, a volatile acid content of ≤1.2 g/L is suitable for a fermented sample, and all the samples met this requirement. The volatile acid contents of the three mixed fermented beverages were significantly higher than that of the control (*p* < 0.05), possibly due to the leaching of acidic substances from the added mulberry leaf powder. In addition, the total SO_2_ content of fermented beverages should not exceed 250 mg/mL, and the four samples were within the acceptable range. Free SO_2_ inhibits the oxidative browning of fermented beverages, and free SO_2_ contents below 3.92 mg/L result in weak antioxidant effects [31]. The lowest free SO_2_ content in the four samples was 4.2 ± 0.20 mg/L. Overall, the oenological parameters of the four samples fall essentially within the allowable ranges of the fruit fermented beverage industry.

As shown in Table 2, the total sugar contents of the four samples decreased significantly between days 0 and 3 (*p* < 0.05), after which the total sugars plateaued between days 3 and 9 (*p* ≥ 0.05). Furthermore, the slow decline in the total sugar content between days 0 and 1 was due to the slow growth rate of yeast, with yeast numbers remaining at low levels. The significant drop in sugar and the logarithmic growth of the yeast between days 1 and 3 represented the conversion of sugar by the yeast into carbon dioxide and alcohol. Between days 3 and 9, the total sugar content tended to be stable as the amount of sugar available to the yeast was low. In addition, the increased alcohol concentrations also affect the growth of the yeast [32]. At the end of fermentation, the total sugar content was reduced to less than 5 mg/mL, indicating that the fermentation endpoint had been reached, and the total sugar contents of the three mixed fermented alcoholic beverages were significantly lower than that of the control.

### 3.2. Analysis of Phenolic Substances and Antioxidant Activity in Mixed Mulberry Fruit and Mulberry Leaf Fermented Alcoholic Beverages

The total phenolic content of the fermented alcoholic beverages is shown in Table 2. The total phenolic content decreased significantly between days 0 and 5, after which it stabilized between days 5 and 9. In addition, the total phenolic contents of the three mixed fermented alcoholic beverages were significantly lower than that of the control between days 3 and 5. However, there were no significant differences in the total phenolic contents of the four samples at the end of fermentation. Chen et al. [33] also found that the total phenolic content in blueberry fermented alcoholic beverages tended to decrease with prolonged fermentation. This decline may be caused by reactions of the active yeast with many secondary metabolites and total phenols to generate derivatives during fermentation.

The contents of total anthocyanins, anthocyanin monomers and anthocyanin polymers are shown in Table 2. After fermentation for 9 days, the total anthocyanin contents of all samples decreased significantly, with no significant differences observed between the four samples. Anthocyanin composition is influenced by a variety of factors, including weather, gene expression levels and anthocyanins in plants. Anabolism of other secondary metabolites may be caused by the reaction of the active yeast with various secondary metabolites and anthocyanins during the fermentation process [34]. In addition, anthocyanins are adsorbed by yeast cell walls, and anthocyanins may also degrade spontaneously [35]. The anthocyanin contents, including total, monomeric and polymeric anthocyanins, also decreased significantly with the prolongation of fermentation time. The contents of monomeric, polymeric and total anthocyanins in the three groups with different concentrations of mulberry leaf powder gradually decreased, with the highest degradation rate seen in the monomeric forms. Generally, as storage time increases, the retention of anthocyanins tends to decrease. The main pigments in fruit fermented beverages include phenolic substances such as anthocyanins and macromolecular pigments, of which the macromolecular pigments are usually more stable than the free anthocyanins. This suggests a preferential degradation of monomeric anthocyanins [36].

The total flavonol and tartaric ester contents are also shown in Table 2. After fermentation for 9 days, both flavonols and tartrates in the control group decreased significantly. However, there was no significant change in the total flavonol content of the fermented beverages containing mulberry leaves compared with those before fermentation (*p* < 0.05). The flavonol content first increased and then decreased during fermentation, which appeared to be related to the addition of mulberry leaves. Mulberry leaves are rich in flavonols, which are constantly dissolved during the fermentation process [37]. Furthermore, at the end of fermentation, there was also an increase in the tartaric ester content in fermented beverages with added mulberry leaves. Specifically, at a 3% concentration of mulberry leaf powder, the tartaric ester content increased from 0.30 ± 0.00 to 0.4 ± 0.02 mg caffeic acid/mL. Compared with the control, the increased tartrate contents of the mixed fermented beverages and the complex changes in the tartaric ester contents during the fermentation process can be related to the continuous dissolution of phenolic components in mulberry leaves, as well as the continuous polymerization and oxidation of phenolic components [38].

The addition of mulberry leaves also had a significant effect on the color of the fermented alcoholic beverages, with both the CI value and Red% gradually decreasing with the longer fermentation while the Yellow%, Blue% and Tint values gradually increased in correspondence with higher mulberry leaf content [39]. In terms of the antioxidant activities of the four mulberry fermented alcoholic beverages, the ABTS^+^ radical-scavenging ability was significantly enhanced, with no significant differences in ABTS^+^ values between the four samples (*p* ≥ 0.05) (Table 2). In addition, the addition of mulberry leaves also improved the FRAP values in the mulberry fermented alcoholic beverages. This was especially noticeable with the addition of 3% mulberry leaves, where the FRAP values were increased from 17.58 ± 1.69 to 23.12 ± 0.96 μmol Fe^2+^/mL (*p* ≥ 0.05) (Table 2), which may be related to the greater amounts of mulberry leaf powder and the dissolution of functional antioxidant components in the mulberry leaves [40].

### 3.3. Determination of Phenolics by HPLC

#### 3.3.1. Changes in the Composition of Anthocyanins

Cyanidin-3-*O*-rutinoside and cyanidin-3-*O*-glucoside were the main anthocyanins found in the mulberry fermented alcoholic beverages [41]. As fermentation progressed, the contents of these two anthocyanins declined significantly between days 0 and 5 and remained stable between days 5 and 9 in the four samples. Reduced concentrations of both cyanidin-3-*O*-rutinoside and cyanidin-3-*O*-glucoside were seen at the end of fermentation. This is due to the large consumption of sugar in the fermentation process of yeast, which accelerates the decomposition of anthocyanin contents. Furthermore, the contents of cyanidin-3-*O*-rutinoside and cyanidin-3-*O*-glucoside were decreased by the addition of mulberry leaf powder (*p* ≥ 0.05) (Table 3), but the contents of cyanidin-3-*O*-rutinoside and cyanidin-3-*O*-glucoside in the 3% concentration group were the lowest at the end of fermentation. The thermal degradation of mulberry anthocyanins is a first-order kinetic reaction. The study of Zhang et al. [42] showed that the degradation of mulberry anthocyanin can be accelerated under high temperature, alkaline conditions and long-term fermentation. This finding was consistent with our results. The pH value of the control group was relatively low, implying the acidic environment of the fermentation. In this case, anthocyanins were liable to be more stable. With the addition of mulberry leaf powder, the pH value increased, and the anthocyanin release rate declined.

#### 3.3.2. Changes in the Composition of Phenolic Acids

Five phenolic acids were detected in the mulberry fermented alcoholic beverages by HPLC-MS. These were protocatechuic acid, caffeic acid, *p*-hydroxybenzoic acid, 4-hydroxycinnamic acid and veratric acid (Table 3). The contents of protocatechuic acid, caffeic acid, *p*-hydroxybenzoic acid and 4-hydroxycinnamic acid all increased during fermentation, while the contents of veratric acid first increased and then decreased until approximating the levels seen before fermentation. Specifically, the contents of caffeic acid, *p*-hydroxybenzoic acid and 4-hydroxycinnamic acid all increased significantly between days 0 and 5 and remained stable between days 5 and 9. However, the content of protocatechuic acid continued to increase during the entire fermentation period. Protocatechuic acid is a metabolite of cyanidin-3-*O*-glucoside, and a large amount of protocatechuic acid was produced as the concentrations of cyanidin-3-*O*-glucoside decreased during fermentation [43]. After 9 days of fermentation, there were no significant differences in the concentrations of protocatechuic acid and 4-hydroxycinnamic acid among the four samples. The concentrations of caffeic acid and *p*-hydroxybenzoic acid increased with the addition of mulberry leaf powder. Wang et al. [44] studied the phenolic substances such as anthocyanins and caffeic acid in fermented beverages by using simulated grape juice for fermentation and found that the phenolic substances greatly improved the color, astringency, and taste of fermented alcoholic beverages. Aromatic compounds such as catechin and gallic acid contribute to the expression of floral and fruity aromas in fermented alcoholic beverages. In addition, the elevated concentrations of caffeic acid in the samples may be due to the dissolution of caffeic acid from mulberry leaves during the fermentation process [45]. Moreover, the complex microbial contributions to the fermented alcoholic beverages should also be considered, and microbial metabolism in fermented alcoholic beverages with the addition of mulberry leaf powder is more complex and will generate phenolic acids, leading to changes in the contents of *p*-hydroxybenzoic acid [46].

#### 3.3.3. Changes in the Flavonol Components

HPLC-MS identified the presence of four flavonols in the mulberry fermented alcoholic beverages, namely rutin, dihydroquercetin, quercetin and myricetin (Table 3). The rutin concentrations in the control and low mulberry leaf groups first increased and then decreased during fermentation. In the 2% concentration group, the rutin content increased gradually between days 0 and 5, after which it remained stable from day 5 to day 9. The rutin content in the 3% concentration group increased gradually during fermentation. The content of dihydroquercetin increased gradually between days 0 and 5 and thereafter remained stable, except in the 3% concentration group where it continued to increase until day 9. The quercetin contents of the four samples increased gradually over the fermentation period. The content of myricetin in the control and 1% concentration groups increased first and then decreased, while increasing gradually in the 2 and 3% concentration groups between days 0 and 9 and thereafter remaining stable. Overall, the contents of flavonoid monomers in the samples increased with increasing amounts of mulberry leaf powder. Mulberry leaves are very rich in rutin and contain a small amount of quercetin. As fermentation progressed, the rutin and quercetin concentrations in the mulberry leaf powder gradually dissolved, and these conditions were also conducive to flavonoid dissolution [47]. Therefore, the contents of rutin and quercetin in the mixed fermented alcoholic beverages were significantly higher than those in the control group and increased in correspondence with increasing additions of mulberry leaf powder.

### 3.4. Changes in DNJ and GABA Contents in Mixed Mulberry Fruit and Mulberry Leaf Fermented Alcoholic Beverages

The DNJ concentrations increased during fermentation in correspondence with the amount of mulberry leaf powder added. At the end of fermentation, there were no significant differences in the DNJ contents among the three samples with added mulberry leaf powder, although all were significantly higher than the control. The values were 149.10 ± 4.64, 153.39 ± 3.98 and 159.51 ± 2.56 mg/L, respectively, according to the order from low to high dosage of mulberry leaf powder (Figure 1). Mulberry leaves fermented with *Lactobacillus plantarum* and *Zygosaccharomyces rouxii* isolated from Korean traditional bean paste have been found to have significantly increased levels of DNJ [48]. Thus, the DNJ in the mulberry leaf powder was continuously dissolved during the fermentation process, so samples with added mulberry leaf powder had higher DNJ contents.

GABA was not detected at all in the control group, while the GABA contents of the mixed fermented beverages first increased and then decreased (Figure 2). On the fifth day of fermentation, the GABA content in the 1% concentration group had decreased to 0, while at the end of fermentation, GABA was only observed in the 2 and 3% concentration groups, with greater GABA levels seen in the latter; the GABA contents in the 2 and 3% concentration groups were 3.40 ± 0.27 and 3.83 ± 0.02 μg/mL, respectively. The initial increase is likely due to the continuous dissolution of GABA in the mulberry leaf powder, while during fermentation, GABA may be utilized by the yeast cells as a nitrogen source, leading to a decrease in the content [49].

### 3.5. Changes in the Free Amino Acid Contents in Mixed Mulberry Fruit and Mulberry Leaf Fermented Alcoholic Beverages

Mulberry leaves are rich in amino acids which are continuously dissolved during the fermentation process. The increased alcohol content is also conducive to amino acid dissolution, providing a rich nitrogen source for the yeast [50]. After being metabolized and released from the yeast, the total free amino acid content in the mixed fermented alcoholic beverages was significantly higher than that in the fermented alcoholic beverages made only from mulberries.

The contents of 17 free amino acids in the mixed mulberry and mulberry leaf fermented alcoholic beverages are shown in Table 4. The concentrations of Gly did not differ between the four samples (*p* < 0.05). Although there were significant differences in the Cys contents between the three mixed fermented alcoholic beverages, the Cys contents were all significantly higher than those in the control group (*p* ≥ 0.05). Pro was highest in the 2 and 3% concentration groups, with the highest levels seen in the 3% concentration group and lowest in the control group [51]. The contents of 14 other amino acids gradually increased with the addition of mulberry leaves. Mulberry leaves are rich in free amino acids, and although these 17 free amino acids are present in different mulberry varieties, their concentrations differ [52]. In addition, the free amino acids in mulberry leaves provide a rich nitrogen source for yeast; this nitrogen is metabolized and released by the yeast, resulting in a higher concentration of free amino acids in fermented alcoholic beverages. Amino acids are important nutrients and contribute significantly to the flavor and aroma of fruit fermented alcoholic beverages [53]. However, amino acids are also precursors of some biogenic amines that are toxic to humans [54]. Thus, although the 3% concentration group had the highest content of free amino acids and nutritional value, there is also a risk of toxic biogenic amine formation.

### 3.6. Analysis of Volatile Substances in the Mixed Mulberry Fruit and Mulberry Leaf Fermented Alcoholic Beverages

The volatile components in the mixed mulberry fruit and mulberry leaf fermented alcoholic beverages were determined by purging and trapping gas mass spectrometry. A total of 33 volatile compounds were identified, including 6 alcohols; 17 esters; 7 aldehydes; 2 ketones; and naphthalene, 2-methyl- (Table 5). Among them, 17 compounds were present in all samples, namely glycerin; 1-propanol, 2-methyl-; 1-butanol, 3-methyl-; 2-octanol; phenylethyl alcohol; 1-butanol, 3-methyl-, acetate; butanoic acid, ethyl ester; hexanoic acid, ethyl ester; octanoic acid, ethyl ester; decanoic acid, ethyl ester; dodecanoic acid, ethyl ester; acetaldehyde; 5-hydroxymethylfurfural; tetradecane; pentadecane; hexadecane; and naphthalene, 2-methyl- [55]. Twenty volatile substances were detected in the control group, 22 in the 2 and 3% concentration groups, and 28 in the 1% concentration group.

The thermal map analysis of the 33 volatile compounds is shown in Figure 3. The aroma of fermented alcoholic beverages is mainly determined by aromatic compounds, including esters, acids, higher alcohols and aldehydes [56].

The aroma relies on multitudes of aromatic compounds with alcohols, esters and aldehydes as the main aroma compounds [57]. The aroma of mulberry fermented alcoholic beverages is mainly apple, grape, coconut and other fruits [58]. Esters are dominant flavor substances, with most having rich aromas of fresh fruit. For example, the presence of decanoic acid, ethyl ester, in the four samples contributed to the fruity flavors of grape and coconut, while the octanoic ethyl ester had fruity, star anise and sweet flavors. Of these, octanoic acid, ethyl ester, has a fruity aroma and is mainly used in the manufacture of condiments and spices [59].

Alcohols are mainly produced by microbial metabolism, unsaturated fatty acid degradation, and reduction reactions of carbonyl compounds in the fermentation process, contributing to the aroma of cloves [60]. 1-Propanol, 2-methyl-; 1-butanol, 3-methyl-; and 2-octanol were the dominant alcohols found in mulberry fermented beverages under investigation. Compared with esters and alcohols, aldehydes and ketones include fewer volatile flavor substances but also contribute to the liquor flavor [61].

In terms of the effect of different amounts of mulberry leaves on the volatility characteristics of the samples, it was found that there were greater varieties and concentrations of alcohols and esters in the 1% concentration group, followed by the 2% concentration group, and the 3% concentration group contained the least, closely resembling the control group. This might be because larger amounts of mulberry leaf powder have greater effects on the yeast, which are not conducive to yeast reproduction or metabolism, thereby reducing the levels of volatile substances.

### 3.7. Principal Component Analysis

Principal component analysis (PCA) was used to analyze the indicators of the samples at two time points (day 5 and day 9) during fermentation (Figure 4). The analysis included total sugars, total phenols, total anthocyanins, total flavonol, tartrate, pH, monomer anthocyanins, polymeric anthocyanins, protocatechuic acid, *p*-hydroxybenzoic acid, caffeic acid, 4-hydroxycinnamic acid, veratric acid, Fe^3+^ reducing powers, ABTS^+^ radical-scavenging ability, cyanidin-3-*O*-glucoside, cyanidin-3-*O*-rutinoside, DNJ and GABA. Three principal components PC1, PC2 and PC3 were extracted, which accounted for 62.6, 16.7 and 13.3%, respectively, of the total variance of the 19-variable system, and the cumulative contribution rate of the three principal components was 92.6%. Two time points in the 3% concentration group and the end of fermentation in the 1 and 2% concentration groups fall on the positive side of PC1 in the score plot, while both time points in the control group fell on the negative side of PC1. The characteristic value of PC1 was 11.90, in which pH, *p*-hydroxybenzoic acid and caffeic acid had larger loading values. The total phenolics, total anthocyanins, monomer anthocyanins, cyanidin-3-*O*-glucoside, cyanidin-3-*O*-rutinoside, veratric acid and polymeric anthocyanins fell on the negative side of PC1, and the remaining 12 indicators were in the PC1-positive region. The eigenvalue of PC2 was 3.18 and the eigenvalue of PC3 was 2.53. Therefore, in terms of the comprehensive analysis of the various indicators, the 3% concentration group showed the best performance.

PCA was also used to analyze the distribution of 17 free amino acids in the four samples on the ninth day of alcoholic fermentation (Figure 5). Three principal components, PC1, PC2 and PC3, were extracted, which accounted for 92.3, 5.9 and 1.8%, respectively, of the total variance of the 17-variable system, and the cumulative contribution rate of the three principal components was 100%. Among them, the 3% concentration group was on the positive side of PC1 and PC2 in the score chart, the control group was on the negative side of PC1, and the other two groups were distributed in the medium regions of PC1. In the loading diagram, Cys was located on the positive sides of PC1 and PC2, Gly and Pro were located on the positive sides of PC1 and PC3, and Ala and the remaining 13 free amino acids were distributed on the positive side of PC1 and near the origin of PC2. Therefore, the free amino acids in the 3% concentration group were mainly distributed on the positive side of PC1, which is consistent with Table 3. Increased addition of mulberry leaf powder thus increased the contents of free amino acids.

### 3.8. Sensory Property

The mixed mulberry fruit and mulberry leaf fermented alcoholic beverages were further analyzed by sensory evaluation. The ruby red color of the fermented alcoholic beverages in the control group was more intense and significantly higher than that seen in the fermented alcoholic beverages with added mulberry leaf powder, which was consistent with the measurements of the red value (Figure 6A). The mixed fermented alcoholic beverages had a stronger leaf flavor, which was consistent with the addition of the mulberry leaf powder. In addition, the chocolate flavor was observed to increase with the addition of mulberry leaf powder.

There were no significant differences in sweetness, acidity, tannin, aftertaste length and alcohol taste among all samples (Figure 6B). The fruit flavor of the mixed fermented alcoholic beverages was noticeably higher than that of mulberry fermented alcoholic beverages, which might be because the mulberry fermented alcoholic beverages had a more complex aroma, more flavor substance content and a stronger fruit flavor after the addition of the mulberry leaves. Usually, the indicator of a fermented alcoholic beverage’s mint flavor may come from the grapes themselves, or from plants such as eucalyptus or menthol planted around the vineyard. For mulberry wine containing mulberry leaf powders, the taste of mulberry leaves should be close to mint, so it is designated as mint flavor. Oak barrels are usually roasted by fire, and some fermented alcoholic beverages aged in oak barrels develop a smoky flavor [62]. However, the fermented alcoholic beverages in this experiment were not aged in oak barrels. The finding of a smoky flavor may be due to the mulberry leaves being dried in a convection oven, which produces a similar taste to smoke. In general, the peppermint and smoky flavors were proportional to the amount of mulberry leaf powder added.

## 4. Conclusions

In this paper, the effects of additions of different amounts of mulberry leaf powder on the quality of mulberry alcoholic beverages were studied. After fermentation for 9 days, no significant differences in the contents of total phenolics, total anthocyanins, protocatechuic acid, 4-hydroxycinnamic acid and ABTS^+^ radical-scavenging ability were observed among the four samples. While there were no significant differences in the DNJ and total flavonol contents in the three mixed fermented alcoholic beverages, they were all higher than those in the counterpart without the addition of mulberry leaves.

The concentrations of tartaric esters, caffeic acid, *p*-hydroxybenzoic acid, veratric acid, flavonol monomers and free amino acids increased with the increase in addition of mulberry leaf powder, while GABA was detectable in the fermented alcoholic beverages added with mulberry leaves with dosage of 2% and 3%. Sensory evaluation showed that both mint and smoky flavors increased with the increase in addition of mulberry leaf powder. Regarding consumer acceptance, fermented alcoholic beverages with a 3% concentration of mulberry leaves are unsuitable for drinking in terms of taste. Overall, although the levels of functional components in the mulberry alcoholic beverage with the 2% addition of mulberry leaves were not the highest, this fermented alcoholic beverage tasted better. Thus, the mulberry fermented alcoholic beverages with the addition of mulberry leaves at 2% is more acceptable to consumers.

## Figures and Tables

**Figure 1 foods-11-03125-f001:**
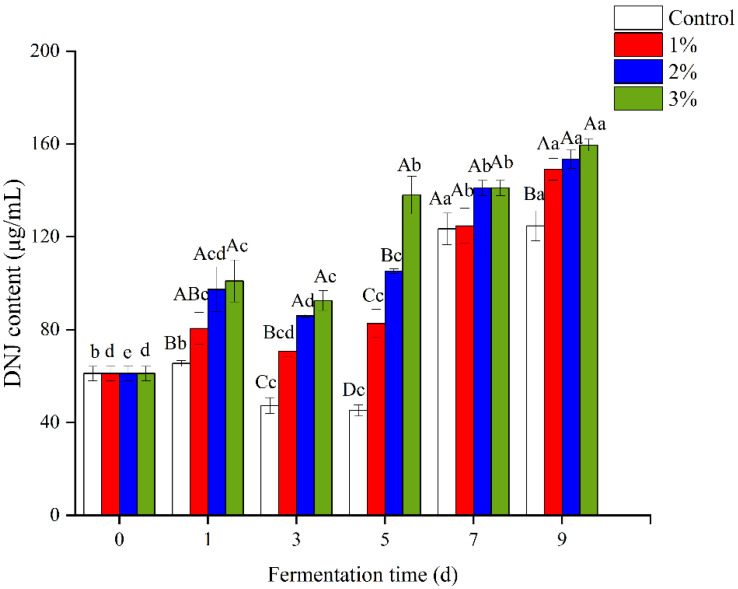
Changes in the 1-deoxynojirimycin (DNJ) content in the fermented alcoholic beverages during fermentation. Values followed by uppercase letters indicate significant differences in added concentration, lowercase letters indicate significant differences in fermentation time (*p* < 0.05). Control represents without the addition of mulberry leaves; 1, 2 and 3% (*w*/*v*, g/mL dry weight) represent the added amount of mulberry leaf powder.

**Figure 2 foods-11-03125-f002:**
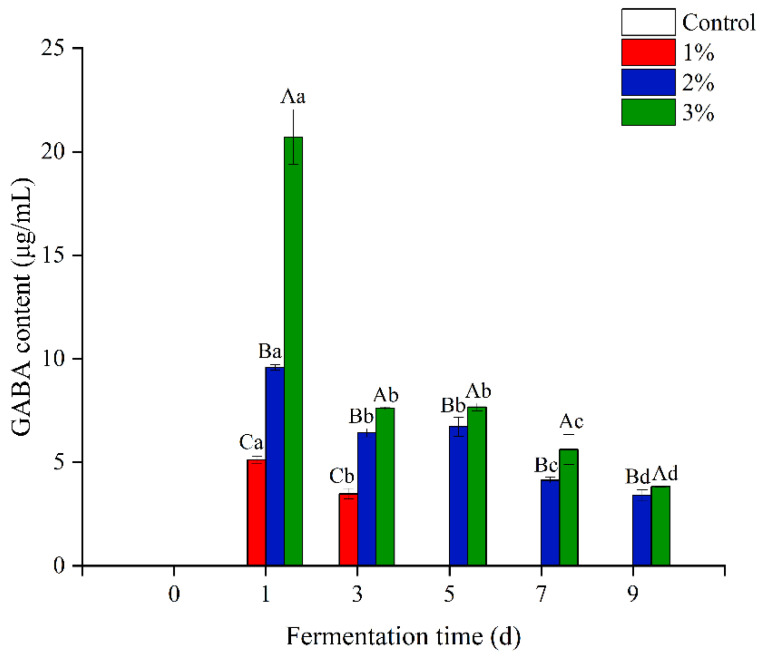
Changes in the γ-aminobutyric acid (GABA) content in the fermented alcoholic beverages with addition of mulberry leaf powder during fermentation. Values followed by uppercase letters indicate significant differences in added concentration, lowercase letters indicate significant differences in fermentation time (*p* < 0.05). Control represents without the addition of mulberry leaves; 1, 2 and 3% (*w*/*v*, g/mL dry weight) represent the added amount of mulberry leaf powder.

**Figure 3 foods-11-03125-f003:**
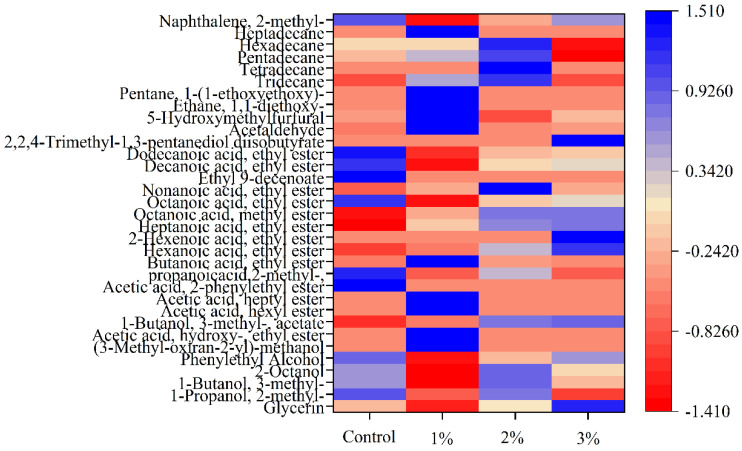
Heatmap showing the relative contents of volatile substances in the four fermented alcoholic beverages with the addition of mulberry leaf powder during fermentation. Control represents without the addition of mulberry leaves; 1, 2 and 3% (*w*/*v*, g/mL dry weight) represent the added amount of mulberry leaf powder.

**Figure 4 foods-11-03125-f004:**
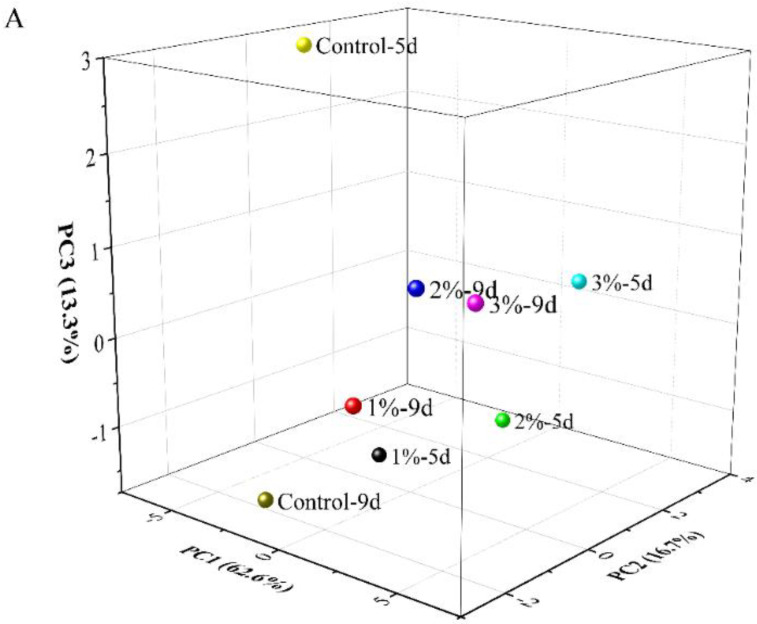
Principal component analysis of the four mulberry fermented alcoholic beverages containing mulberry leaf powder during fermentation. Score plots (**A**) and loading plots (**B**). Control represents without the addition of mulberry leaves; 1, 2 and 3% (*w*/*v*, g/mL dry weight) represent the added amount of mulberry leaf powder.

**Figure 5 foods-11-03125-f005:**
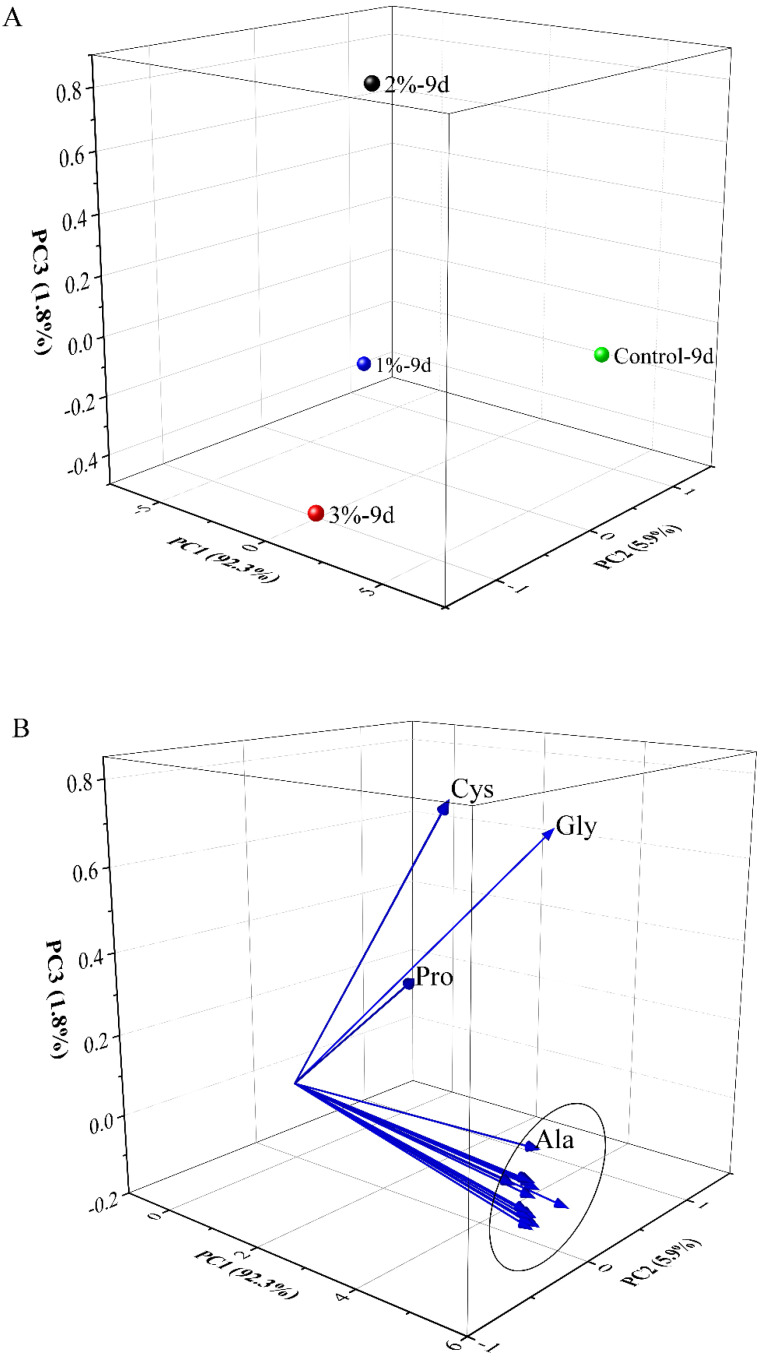
Principal component analysis of free amino acids in the four mulberry fermented alcoholic beverages containing mulberry leaf powder during fermentation. Score plots (**A**) and loading plots (**B**). The free amino acids detected in the four kinds of mulberry fermented beverages were analyzed by PCA using Origin 2021 software. Control represents without the addition of mulberry leaves; 1, 2 and 3% (*w*/*v*, g/mL dry weight) represent the added amount of mulberry leaf powder.

**Figure 6 foods-11-03125-f006:**
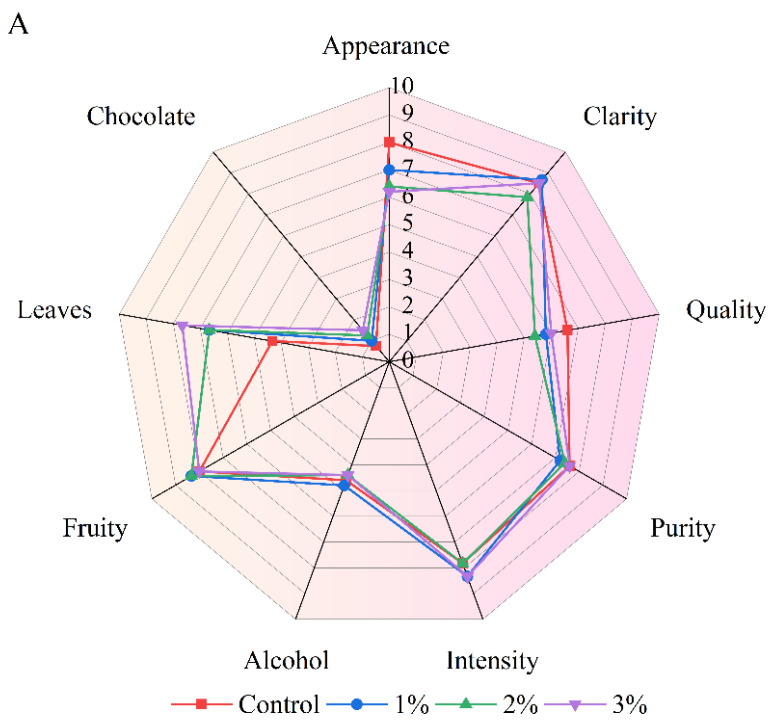
The appearance and aroma scores (**A**) and taste scores (**B**) of the four mulberry alcoholic beverages containing mulberry leaf powder after fermentation. Control represents without the addition of mulberry leaves; 1, 2 and 3% (*w*/*v*, g/mL dry weight) represent the added amount of mulberry leaf powder.

**Table 1 foods-11-03125-t001:** Basic oenological parameters of mulberry fermented alcoholic beverages containing different amounts of mulberry leaf powders (on Day 9).

Parameters	Control	1%	2%	3%
Alcohol (%)	12.20 ± 0.81 ^a^	12.20 ± 0.46 ^a^	12.80 ± 0.37 ^a^	12.20 ± 0.63 ^a^
Volatile acid (g acetic acid/L)	0.49 ± 0.11 ^b^	1.03 ± 0.24 ^a^	0.95 ± 0.12 ^a^	1.00 ± 0.38 ^a^
Total SO_2_ (mg/L)	15.00 ± 1.00 ^c^	22.00 ± 1.56 ^a^	18.00 ± 1.25 ^b^	18.00 ± 1.06 ^b^
Free SO_2_ (mg/L)	4.20 ± 0.20 ^c^	7.01 ± 0.41 ^a^	7.01 ± 0.58 ^a^	5.61 ± 0.32 ^b^
pH	3.90 ± 0.02 ^d^	4.04 ± 0.01 ^c^	4.10 ± 0.01 ^b^	4.17 ± 0.01 ^a^

Data represent mean values of three replicates ± standard deviation. Values followed by different letters in each row indicate significant differences (*p* < 0.05). Control represents without the addition of mulberry leaves; 1, 2 and 3% (*w*/*v*, g/mL dry weight) represent the added amount of mulberry leaf powders.

**Table 2 foods-11-03125-t002:** Total sugar content, phenolic contents, color and antioxidant activities of mixed mulberry fruit and mulberry leaf fermented alcoholic beverages.

Parameter	Additive Amount	Time (Day)					
		0	1	3	5	7	9
Total sugar (mg/mL)	Control	132.19 ± 12.00 ^a^	77.48 ± 4.52 ^Bb^	9.74 ± 0.06 ^Ac^	3.74 ± 0.05 ^Bc^	3.93 ± 0.27 ^Bc^	3.34 ± 0.17 ^Bc^
	1%	132.19 ± 12.00 ^a^	60.97 ± 2.55 ^Cb^	5.15 ± 0.35 ^Cc^	3.54 ± 0.04 ^Bc^	3.78 ± 0.11 ^Bc^	4.52 ± 0.30 ^Ac^
	2%	132.19 ± 12.00 ^a^	66.78 ± 3.54 ^Cb^	5.71 ± 0.16 ^BCc^	4.64 ± 0.21 ^Ac^	3.87 ± 0.18 ^Bc^	4.52 ± 0.27 ^Ac^
	3%	132.19 ± 12.00 ^a^	103.70 ± 2.96 ^Ab^	6.27 ± 0.30 ^Bc^	4.12 ± 0.08 ^Ac^	5.20 ± 0.2 ^Ac^	5.08 ± 0.01 ^Ac^
**Phenolic family**							
Total phenolics (mg GAE/g DW)	Control	1.78 ± 0.01 ^a^	1.73 ± 0.05 ^Aab^	1.69 ± 0.01 ^Abc^	1.64 ± 0.01 ^Ac^	1.62 ± 0.01 ^Ac^	1.62 ± 0.03 ^Ac^
	1%	1.78 ± 0.01 ^a^	1.73 ± 0.02 ^Ab^	1.61 ± 0.02 ^Bc^	1.61 ± 0.01 ^Bc^	1.61 ± 0.01 ^Ac^	1.62 ± 0.01 ^Ac^
	2%	1.78 ± 0.01 ^a^	1.74 ± 0.01 ^Aa^	1.63 ± 0.02 ^Bb^	1.60 ± 0.00 ^Bb^	1.60 ± 0.01 ^Ab^	1.60 ± 0.03 ^Ab^
	3%	1.78 ± 0.01 ^a^	1.75 ± 0.03 ^Aa^	1.65 ± 0.01 ^Bb^	1.59 ± 0.05 ^Bc^	1.60 ± 0.05 ^Abc^	1.61 ± 0.02 ^Abc^
Total anthocyanins (mg/mL)	Control	1.14 ± 0.00 ^a^	0.70 ± 0.02 ^Ab^	0.52 ± 0.03 ^Ac^	0.46 ± 0.03 ^Ad^	0.39 ± 0.00 ^Ae^	0.42 ± 0.03 ^Ae^
	1%	1.14 ± 0.00 ^a^	0.55 ± 0.04 ^Bb^	0.44 ± 0.00 ^Ac^	0.42 ± 0.02 ^ABc^	0.33 ± 0.01 ^Bd^	0.37 ± 0.01 ^Ad^
	2%	1.14 ± 0.00 ^a^	0.55 ± 0.00 ^Bb^	0.41 ± 0.06 ^Ac^	0.38 ± 0.01 ^Bcd^	0.31 ± 0.00 ^Bcd^	0.37 ± 0.00 ^Ad^
	3%	1.14 ± 0.00 ^a^	0.60 ± 0.04 ^Bb^	0.41 ± 0.01 ^Ac^	0.36 ± 0.02 ^Bcd^	0.33 ± 0.01 ^Bd^	0.33 ± 0.02 ^Ad^
Monomeric anthocyanins (AU)	Control	17.45 ± 0.37 ^a^	13.23 ± 0.11 ^Ab^	8.43 ± 0.14 ^Ac^	7.1 ± 0.03 ^Ad^	6.64 ± 0.17 ^Ad^	6.67 ± 0.11 ^Ad^
	1%	17.45 ± 0.37 ^a^	9.04 ± 0.46 ^Cb^	6.67 ± 0.25 ^Bc^	5.95 ± 0.25 ^Bc^	5.11 ± 0.25 ^Bd^	5.02 ± 0.04 ^Bd^
	2%	17.45 ± 0.37 ^a^	10.35 ± 0.25 ^BCb^	4.98 ± 0.45 ^Cc^	4.29 ± 0.23 ^Cc^	4.08 ± 0.06 ^Cc^	4.09 ± 0.01 ^Cc^
	3%	17.45 ± 0.37 ^a^	11.25 ± 0.84 ^Bb^	4.25 ± 0.25 ^Cc^	3.69 ± 0.04 ^Dc^	4.35 ± 0.17 ^Cc^	4.14 ± 0.23 ^Cc^
Polymeric anthocyanins (AU)	Control	3.69 ± 0.13 ^a^	3.53 ± 0.08 ^Aa^	3.24 ± 0.18 ^Aab^	3.31 ± 0.21 ^Aab^	2.97 ± 0.10 ^Ab^	2.91 ± 0.04 ^Ab^
	1%	3.69 ± 0.13 ^a^	2.64 ± 0.20 ^Bb^	2.43 ± 0.05 ^Bb^	2.53 ± 0.02 ^Bb^	1.92 ± 0.01 ^Bc^	1.95 ± 0.16 ^Bc^
	2%	3.69 ± 0.13 ^a^	2.58 ± 0.22 ^Bb^	2.10 ± 0.04 ^Bc^	2.10 ± 0.20 ^Cc^	1.57 ± 0.02 ^Cd^	1.61 ± 0.02 ^Cd^
	3%	3.69 ± 0.13 ^a^	2.65 ± 0.25 ^Bb^	2.16 ± 0.15 ^Bc^	1.90 ± 0.01 ^Cc^	1.52 ± 0.04 ^Cd^	1.4 ± 0.01 ^Cd^
Total anthocyanins (AU)	Control	20.2 ± 1.22 ^a^	16.61 ± 0.83 ^Ab^	11.28 ± 0.04 ^Ac^	10.21 ± 0.01 ^Acd^	9.32 ± 0.17 ^Ad^	8.36 ± 0.09 ^Ad^
	1%	20.2 ± 1.22 ^a^	11.62 ± 0.59 ^Bb^	8.84 ± 0.44 ^Bc^	8.07 ± 0.21 ^Bcd^	6.98 ± 0.25 ^Bd^	6.29 ± 0.28 ^Bd^
	2%	20.2 ± 1.22 ^a^	11.70 ± 0.76 ^Bb^	6.725 ± 0.28 ^Cc^	6.12 ± 0.12 ^Cc^	5.22 ± 0.36 ^Cc^	5.62 ± 0.08 ^Cc^
	3%	20.2 ± 1.22 ^a^	13.15 ± 0.69 ^Bb^	6.17 ± 0.50 ^Cc^	5.49 ± 0.03 ^Dc^	5.56 ± 0.26 ^Cc^	5.46 ± 0.24 ^Cc^
Total flavonols (mg RE/g DW)	Control	0.23 ± 0.00 ^a^	0.22 ± 0.02 ^Ba^	0.23 ± 0.00 ^Ba^	0.22 ± 0.01 ^Ba^	0.21 ± 0.00 ^Ba^	0.13 ± 0.01 ^Bb^
	1%	0.23 ± 0.00 ^b^	0.27 ± 0.01 ^Aa^	0.25 ± 0.01 ^Bb^	0.16 ± 0.00 ^Cd^	0.20 ± 0.00 ^Cc^	0.23 ± 0.01 ^Ab^
	2%	0.23 ± 0.00 ^bc^	0.29 ± 0.01 ^Aa^	0.26 ± 0.02 ^Bb^	0.17 ± 0.01 ^Cd^	0.21 ± 0.00 ^BCc^	0.24 ± 0.01 ^Ab^
	3%	0.23 ± 0.00 ^c^	0.29 ± 0.02 ^Aab^	0.31 ± 0.02 ^Aa^	0.25 ± 0.00 ^Abc^	0.24 ± 0.00 ^Ac^	0.25 ± 0.02 ^Abc^
Tartaric esters (mg caffeic acid/mL)	Control	0.30 ± 0.00 ^a^	0.29 ± 0.01 ^Ba^	0.30 ± 0.00 ^Ca^	0.29 ± 0.01 ^Ba^	0.30 ± 0.00 ^Da^	0.22 ± 0.02 ^Cb^
	1%	0.30 ± 0.00 ^c^	0.35 ± 0.00 ^ABa^	0.36 ± 0.02 ^Ba^	0.23 ± 0.00 ^Cd^	0.32 ± 0.01 ^Cb^	0.29 ± 0.01 ^Bc^
	2%	0.30 ± 0.00 ^b^	0.40 ± 0.03 ^Aa^	0.37 ± 0.01 ^Ba^	0.27 ± 0.00 ^Bb^	0.35 ± 0.00 ^Ba^	0.36 ± 0.02 ^Aa^
	3%	0.30 ± 0.00 ^b^	0.41 ± 0.03 ^Aa^	0.47 ± 0.04 ^Aa^	0.39 ± 0.01 ^Aa^	0.41 ± 0.01 ^Aa^	0.40 ± 0.02 ^Aa^
**Color**							
CI	Control	3.31 ± 0.33 ^a^	2.92 ± 0.14 ^Aa^	2.17 ± 0.08 ^Ab^	2.17 ± 0.23 ^Ab^	2.03 ± 0.03 ^Ab^	1.99 ± 0.04 ^Ab^
	1%	3.31 ± 0.33 ^a^	2.16 ± 004 ^Bb^	1.71 ± 0.09 ^Bc^	1.79 ± 0.03 ^Bc^	1.48 ± 0.07 ^Bc^	1.48 ± 0.02 ^Bc^
	2%	3.31 ± 0.33 ^a^	2.15 ± 0.15 ^Bb^	1.42 ± 0.12 ^Bc^	1.42 ± 0.03 ^Cc^	1.33 ± 0.05 ^BCc^	1.42 ± 0.00 ^Cc^
	3%	3.31 ± 0.33 ^a^	2.67 ± 0.13 ^Ab^	1.40 ± 0.02 ^Bc^	1.36 ± 0.00 ^Cc^	1.21 ± 0.08 ^Cc^	1.14 ± 0.00 ^Dc^
Red%	Control	60.41 ± 1.00 ^a^	57.98 ± 0.30 ^ABb^	54.40 ± 0.l4 ^Ac^	53.31 ± 0.51 ^Acd^	52.3 5± 0.40 ^Ade^	51.15 ± 0.57 ^Ae^
	1%	60.41 ± 1.00 ^a^	58.62 ± 0.03 ^Ab^	52.49 ± 0.99 ^Bc^	50.92 ± 0.26 ^Bc^	51.08 ± 0.14 ^Bc^	50.56 ± 0.17 ^Ac^
	2%	60.41 ± 1.00 ^a^	57.01 ± 0.57 ^Bb^	51.70 ± 0.45 ^BCc^	49.19 ± 0.43 ^Cd^	48.82 ± 0.04 ^Cd^	49.28 ± 0.17 ^Bd^
	3%	60.41 ± 1.00 ^a^	57.31 ± 0.36 ^ABb^	50.18 ± 0.05 ^Cc^	47.96 ± 0.14 ^Dd^	48.66 ± 0.12 ^Cd^	47.72 ± 0.12 ^Cd^
Yellow%	Control	31.93 ± 0.27 ^f^	32.35 ± 0.10 ^Be^	36.89 ± 0.04 ^Cd^	37.99 ± 0.08 ^Dc^	38.79 ± 0.24 ^Db^	39.16 ± 0.14 ^Da^
	1%	31.93 ± 0.27 ^d^	33.62 ± 0.06 ^Ac^	38.87 ± 0.93 ^Bb^	39.69 ± 0.28 ^Cab^	40.09 ± 0.15 ^Cab^	40.44 ± 0.27 ^Ca^
	2%	31.93 ± 0.27 ^e^	33.77 ± 0.07 ^Ad^	39.43 ± 0.09 ^ABc^	40.91 ± 0.11 ^Bb^	41.51 ± 0.02 ^Ba^	41.69 ± 0.00 ^Ba^
	3%	31.93 ± 0.27 ^d^	33.75 ± 0.55 ^Ac^	40.59 ± 0.21 ^Ab^	42.83 ± 0.05 ^Aa^	43.04 ± 0.67 ^Aa^	42.95 ± 0.19 ^Aa^
Blue%	Control	7.66 ± 0.73 ^b^	9.66 ± 0.40 ^Aa^	8.71 ± 0.41 ^Aab^	8.70 ± 0.43 ^Aab^	8.91 ± 0.16 ^ABab^	9.70 ± 0.43 ^Aa^
	1%	7.66 ± 0.73 ^b^	7.76 ± 0.03 ^Bb^	8.64 ± 0.06 ^Aa^	9.40 ± 0.02 ^Aa^	8.83 ± 0.01 ^ABa^	9.00 ± 0.10 ^Aa^
	2%	7.66 ± 0.73 ^b^	9.22 ± 0.65 ^Aab^	8.87 ± 0.36 ^Aab^	9.90 ± 0.54 ^Aa^	9.67 ± 0.06 ^Aa^	9.04 ± 0.17 ^Aab^
	3%	7.66 ± 0.73 ^b^	8.94 ± 0.20 ^Aa^	9.23 ± 0.15 ^Aa^	9.20 ± 0.09 ^Aa^	8.30 ± 0.55 ^Bab^	9.33 ± 0.31 ^Aa^
Tint	Control	0.53 ± 0.01 ^f^	0.56 ± 0.00 ^Be^	0.68 ± 0.00 ^Cd^	0.71 ± 0.01 ^Dc^	0.74 ± 0.01 ^Db^	0.77 ± 0.01 ^Da^
	1%	0.53 ± 0.01 ^d^	0.57 ± 0.00 ^ABc^	0.74 ± 0.03 ^Bb^	0.78 ± 0.01 ^Cab^	0.78 ± 0.01 ^Cab^	0.80 ± 0.01 ^Ca^
	2%	0.53 ± 0.01 ^d^	0.59 ± 0.00 ^Ac^	0.76 ± 0.01 ^ABb^	0.83 ± 0.01 ^Ba^	0.85 ± 0.00 ^Ba^	0.85 ± 0.00 ^Ba^
	3%	0.53 ± 0.01 ^d^	0.59 ± 0.01^Ac^	0.81 ± 0.00 ^Ab^	0.89 ± 0.00 ^Aa^	0.88 ± 0.02 ^Aa^	0.90 ± 0.00 ^Aa^
**Antioxidant activity**							
ABTS^+^ (μmol Trolox/L)	Control	155.36 ± 11.81 ^b^	206.04 ± 0.84 ^Aa^	220.95 ± 5.06 ^Aa^	228.70 ± 14.33 ^Aa^	228.11 ± 1.69 ^Aa^	229.90 ± 5.90 ^Aa^
	1%	155.36 ± 11.81 ^d^	210.22 ± 1.69 ^Ac^	212.01 ± 0.84 ^Ac^	223.93 ± 2.53 ^Abc^	235.86 ± 4.22 ^Ab^	255.54 ± 8.43 ^Aa^
	2%	155.36 ± 11.81 ^d^	209.02 ± 13.49 ^Ac^	214.99 ± 8.43 ^Abc^	238.84 ± 1.69 ^Aab^	243.01 ± 2.53 ^Aab^	256.13 ± 12.65 ^Aa^
	3%	155.36 ± 11.81 ^c^	218.57 ± 1.69 ^Ab^	228.11 ± 11.81 ^Ab^	241.82 ± 7.59 ^Aab^	242.42 ± 11.81 ^Aab^	262.69 ± 6.75 ^Aa^
FRAP (μmol Fe^2+^/mL)	Control	17.58 ± 1.69 ^c^	31.67 ± 1.86 ^Aa^	25.76 ± 1.52 ^Ab^	25.84 ± 1.19 ^Ab^	21.73 ± 1.24 ^ABbc^	19.65 ± 1.02 ^Bc^
	1%	17.58 ± 1.69 ^b^	24.73 ± 1.75 ^Ba^	19.10 ± 1.47 ^Bb^	18.70 ± 1.36 ^Bb^	17.46 ± 1.75 ^Bb^	18.86 ± 0.68 ^Bb^
	2%	17.58 ± 1.69 ^b^	22.45 ± 1.13 ^Ba^	23.69 ± 0.40 ^Aa^	20.57 ± 1.98 ^ABab^	20.65 ± 0.62 ^ABab^	20.41 ± 0.17 ^Bab^
	3%	17.58 ± 1.69 ^b^	22.97 ± 0.17 ^Ba^	26.44 ± 1.58 ^Aa^	26.48 ± 1.98 ^Aa^	25.05 ± 2.42 ^Aa^	23.12 ± 0.96 ^Aa^

Data represent mean values of three replicates ± standard deviation. Values followed by uppercase letters indicate significant differences in added concentration, lowercase letters indicate significant differences in fermentation time (*p* < 0.05). Control represents without the addition of mulberry leaves; 1, 2 and 3% (*w*/*v*, g/mL dry weight) represent the added amount of mulberry leaf powder.

**Table 3 foods-11-03125-t003:** Contents (mg/L) of individual phenolic compounds in mixed mulberry fruit and mulberry leaf fermented alcoholic beverages.

Parameter	Additive Amount	Time (Day)		
		0	5	9
**Anthocyanins**				
Cyanidin-3-*O*-glucoside (mg/L)	Control	548.71 ± 43.20 ^a^	78.97 ± 0.86 ^Ab^	50.18 ± 0.38 ^Ab^
	1%	548.71 ± 43.20 ^a^	49.26 ± 2.77 ^Bb^	30.25 ± 0.41 ^Bb^
	2%	548.71 ± 43.20 ^a^	42.46 ± 3.61 ^Cb^	25.56 ± 2.31 ^Cb^
	3%	548.71 ± 43.20 ^a^	33.15 ± 1.07 ^Db^	18.35 ± 0.26 ^Db^
Cyanidin-3-*O*-rutinoside (mg/L)	Control	617.03 ± 28.24 ^a^	370.30 ± 6.02 ^Ab^	322.76 ± 4.43 ^Ab^
	1%	617.03 ± 28.24 ^a^	305.34 ± 26.62 ^Bb^	258.30 ± 9.50 ^Bb^
	2%	617.03 ± 28.24 ^a^	296.79 ± 18.81 ^Bb^	253.57 ± 12.58 ^Bb^
	3%	617.03 ± 28.24 ^a^	258.06 ± 4.11 ^Bb^	236.44 ± 1.72 ^Bb^
**Phenolic acids**				
Protocatechuic acid (mg/L)	Control	17.95 ± 0.27 ^c^	43.17 ± 2.15 ^Bb^	73.33 ± 1.17 ^Aa^
	1%	17.95 ± 0.27 ^c^	45.14 ± 0.48 ^Bb^	71.35 ± 7.00 ^Aa^
	2%	17.95 ± 0.27 ^c^	44.36 ± 4.35 ^Bb^	74.87 ± 2.79 ^Aa^
	3%	17.95 ± 0.27 ^c^	58.78 ± 0.25 ^Ab^	77.26 ± 1.81 ^Aa^
*p*-Hydroxybenzoic acid (mg/L)	Control	7.14 ± 0.62 ^c^	15.50 ± 0.58 ^Cb^	24.15 ± 0.17 ^Ba^
	1%	7.14 ± 0.62 ^c^	24.54 ± 0.54 ^Bb^	30.37 ± 1.78 ^Aa^
	2%	7.14 ± 0.62 ^c^	25.61 ± 1.05 ^Bb^	32.04 ± 1.90 ^Aa^
	3%	7.14 ± 0.62 ^c^	33.53 ± 0.90 ^Aa^	32.88 ± 0.41 ^Aa^
Caffeic acid (mg/L)	Control	3.09 ± 0.08 ^b^	20.02 ± 0.69 ^Da^	19.22 ± 0.37 ^Da^
	1%	3.09 ± 0.08 ^b^	42.42 ± 1.43 ^Ca^	44.38 ± 2.31 ^Ca^
	2%	3.09 ± 0.08 ^b^	51.98 ± 2.42 ^Bb^	58.94 ± 1.64 ^Ba^
	3%	3.09 ± 0.08 ^b^	65.09 ± 3.86 ^Aa^	66.81 ± 2.51 ^Aa^
4-Hydroxycinnamic acid (mg/L)	Control	5.36 ± 0.36 ^c^	15.09 ± 0.34 ^Ab^	16.91 ± 0.37 ^Aa^
	1%	5.36 ± 0.36 ^b^	16.86 ± 0.42 ^Aa^	16.47 ± 1.01 ^Aa^
	2%	5.36 ± 0.36 ^b^	16.84 ± 1.20 ^Aa^	15.21 ± 1.24 ^Aa^
	3%	5.36 ± 0.36 ^b^	16.95 ± 1.48 ^Aa^	16.45± 0.44 ^Aa^
Veratric acid (mg/L)	Control	2.26 ± 0.22 ^b^	5.38 ± 0.22 ^Ba^	2.52 ± 0.03 ^Bb^
	1%	2.26 ± 0.22 ^b^	6.54 ± 0.59 ^Aa^	2.89 ± 0.11 ^Ab^
	2%	2.26 ± 0.22 ^c^	6.92 ± 0.16 ^Aa^	2.93 ± 0.13 ^Ab^
	3%	2.26 ± 0.22 ^b^	7.84 ± 0.51 ^Aa^	3.08 ± 0.08 ^Ab^
**Flavonols**				
Rutin (mg/L)	Control	3.82 ± 0.17 ^b^	6.07 ± 0.28 ^Da^	3.21 ± 0.04 ^Dc^
	1%	3.82 ± 0.17 ^c^	18.70 ± 0.22 ^Ca^	14.56 ± 0.87 ^Cb^
	2%	3.82 ± 0.17 ^b^	25.98 ± 0.63 ^Ba^	26.46 ± 1.18 ^Ba^
	3%	3.82 ± 0.17 ^c^	33.24 ± 1.61 ^Ab^	39.10 ± 2.16 ^Aa^
Myricetin (mg/L)	Control	0.29 ± 0.03 ^b^	0.38 ± 0.00 ^Da^	0.15 ± 0.01 ^Dc^
	1%	0.29 ± 0.03 ^c^	3.52 ± 0.00 ^Ca^	2.26 ± 0.20 ^Cb^
	2%	0.29 ± 0.03 ^b^	5.90 ± 0.53 ^Ba^	5.36 ± 0.06 ^Ba^
	3%	0.29 ± 0.03 ^b^	8.85 ± 0.33 ^Aa^	9.02 ± 0.37 ^Aa^
Quercetin (mg/L)	Control	0.45 ± 0.01 ^c^	2.74 ± 0.14 ^Db^	3.47 ± 0.04 ^Ca^
	1%	0.45 ± 0.01 ^b^	9.55 ± 0.23 ^Ca^	10.37 ± 0.53 ^Ba^
	2%	0.45 ± 0.01 ^c^	14.63 ± 0.37 ^Bb^	20.14 ± 0.08 ^Aa^
	3%	0.45 ± 0.01 ^c^	17.45 ± 0.96 ^Ab^	22.39 ± 2.03 ^Aa^
Dihydroquercetin (mg/L)	Control	7.15 ± 0.01 ^b^	24.70 ± 0.77 ^Ba^	26.76 ± 1.49 ^Ca^
	1%	7.15 ± 0.01 ^b^	30.74 ± 2.82 ^Aa^	33.44 ± 1.61 ^Ba^
	2%	7.15 ± 0.01 ^b^	32.38 ± 0.20 ^Aa^	33.31 ± 1.43 ^Ba^
	3%	7.15 ± 0.01 ^c^	34.60 ± 1.43 ^Ab^	39.30 ± 0.72 ^Aa^

Data are mean value of three replicates ± standard deviation. Values followed by uppercase letters indicate significant differences in added concentration, lowercase letters indicate significant differences in fermentation time (*p* < 0.05). Control represents without the addition of mulberry leaves; 1, 2 and 3% (*w*/*v*, g/mL dry weight) represent the added amount of mulberry leaf powder.

**Table 4 foods-11-03125-t004:** Contents (mg/L) of free amino acids in the four mulberry fermented alcoholic beverages with different amounts of mulberry leaf powder after fermentation.

Amino Acid Species	Control (mg/L)	1% (mg/L)	2% (mg/L)	3% (mg/L)
Asp	6.30 ± 0.51 ^d^	16.61 ± 1.21 ^c^	22.81 ± 2.01 ^b^	30.71 ± 2.45 ^a^
Thr	6.47 ± 0.62 ^d^	14.62 ± 1.24 ^c^	19.15 ± 1.28 ^b^	36.13 ± 3.12 ^a^
Ser	6.13 ± 0.54 ^d^	12.42 ± 0.76 ^c^	16.43 ± 1.56 ^b^	22.47 ± 1.80 ^a^
Glu	41.87 ± 3.46 ^d^	59.87 ± 4.83 ^c^	71.18 ± 4.68 ^b^	86.18 ± 6.45 ^a^
Gly	28.90 ± 2.72 ^a^	30.18 ± 2.58 ^a^	26.95 ± 2.13 ^a^	33.59 ± 2.67 ^a^
Ala	24.25 ± 2.35 ^c^	40.44 ± 3.45 ^b^	46.34 ± 3.67 ^b^	59.48 ± 3.26 ^a^
Cys	21.03 ± 2.06 ^b^	36.88 ± 2.68 ^a^	35.74 ± 2.89 ^a^	37.99 ± 3.12 ^a^
Val	9.17 ± 0.83 ^d^	17.56 ± 1.21 ^c^	21.75 ± 1.56 ^b^	26.00 ± 1.68 ^a^
Met	3.50 ± 0.34 ^d^	8.02 ± 0.56 ^c^	10.16 ± 0.98 ^b^	13.63 ± 1.2 ^a^
Ile	3.23 ± 0.32 ^d^	6.61 ± 0.47 ^c^	8.27 ± 0.68 ^b^	11.19 ± 1.04 ^a^
Leu	10.24 ± 0.99 ^d^	26.65 ± 2.12 ^c^	34.55 ± 3.02 ^b^	47.48 ± 3.87 ^a^
Tyr	3.75 ± 0.36 ^d^	8.78 ± 0.68 ^c^	11.42 ± 1.02 ^b^	15.46 ± 1.32 ^a^
Phe	24.59 ± 2.31 ^d^	45.17 ± 3.45 ^c^	54.90 ± 2.65 ^b^	69.77 ± 3.67 ^a^
Lys	13.09 ± 1.21 ^d^	33.55 ± 3.02 ^c^	45.79 ± 2.03 ^b^	63.37 ± 3.89 ^a^
His	4.24 ± 0.32 ^d^	9.85 ± 0.63 ^c^	13.48 ± 1.69 ^b^	18.12 ± 1.03 ^a^
Arg	16.25 ± 1.57 ^d^	42.88 ± 2.89 ^c^	58.29 ± 3.45 ^b^	77.66 ± 4.68 ^a^
Pro	11.62 ± 0.98 ^c^	65.00 ± 1.21 ^b^	75.41 ± 2.65 ^a^	71.17 ± 3.24 ^a^
Total	234.63 ± 21.13 ^d^	475.07 ± 32.99 ^c^	572.61 ± 37.95 ^b^	720.37 ± 48.30 ^a^

Data represent means of three replicates ± standard deviation. Values followed by different letters in each row indicate significant differences (*p* < 0.05). Control represents without the addition of mulberry leaves; 1, 2 and 3% (*w*/*v*, g/mL dry weight) represent the added amount of mulberry leaf powder.

**Table 5 foods-11-03125-t005:** Relative amounts of volatile compounds in the four mulberry fermented alcoholic beverages with different amounts of mulberry leaf powder after fermentation.

No.	Volatile Compound	Control	1%	2%	3%
	**Alcohols**				
1	Glycerin	(2.44 ± 0.06) × 10 ^7,c^	(2.57 ± 0.07) × 10 ^7,c^	(4.74 ± 0.38) × 10 ^7,b^	(5.61 ± 0.38) × 10 ^7,a^
2	1-Propanol, 2-methyl-	(5.15 ± 0.12) × 10 ^8,b^	(5.16 ± 0.12) × 10 ^8,b^	(7.27 ± 0.65) × 10 ^8,a^	(3.34 ± 0.20) × 10 ^8,c^
3	1-Butanol, 3-methyl-	(1.07 ± 0.08) × 10 ^9,b^	(1.63 ± 0.11) × 10 ^9,a^	(1.87 ± 0.06) × 10 ^9,a^	(1.19 ± 0.11) × 10 ^9,b^
4	2-Octanol	(4.80 ± 0.08) × 10 ^6,c^	(6.80 ± 0.65) × 10 ^6,ab^	(7.53 ± 0.70) × 10 ^6,a^	(5.68 ± 0.12) × 10 ^6,bc^
5	Phenylethyl alcohol	(1.09 ± 0.06) × 10 ^7,b^	(1.07 ± 0.09) × 10 ^7,b^	(1.25 ± 0.03) × 10 ^7,b^	(1.06 ± 0.13) × 10 ^7,a^
6	(3-Methyl-oxiran-2-yl)-methanol	ND	(1.63 ± 0.12) × 10 ^9^	ND	ND
	Total	1.63 × 10 ^9^	3.82 × 10 ^9^	2.66 × 10 ^9^	1.60 × 10 ^9^
	**Esters**				
7	Acetic acid, hydroxy-, ethyl ester	ND	(2.73 ± 0.10) × 10 ^6^	ND	ND
8	1-Butanol, 3-methyl-, acetate	(3.24 ± 0.19) × 10 ^8,c^	(1.01 ± 0.04) × 10 ^9,a^	(9.87 ± 0.22) × 10 ^8,a^	(8.17 ± 0.62) × 10 ^8,b^
9	Acetic acid, hexyl ester	ND	(5.34 ± 0.07) × 10 ^6^	ND	ND
10	Acetic acid, heptyl ester	ND	(5.27 ± 0.29) × 10 ^6^	ND	ND
11	Acetic acid, 2-phenylethyl ester	(8.88 ± 0.05) × 10 ^6^	ND	ND	ND
12	Propanoic acid, 2-methyl-, 1-(1,1-dimethylethyl)-2-methyl-1,3-propanediyl ester	(8.51 ± 0.23) × 10 ^6,a^	ND	(7.29 ± 0.08) × 10 ^6,b^	ND
13	Butanoic acid, ethyl ester	(5.46 ± 0.08) × 10 ^7,d^	(2.36 ± 0.08) × 10 ^8,a^	(1.09 ± 0.01) × 10 ^8,b^	(7.72 ± 0.45) × 10 ^7,c^
14	Hexanoic acid, ethyl ester	(1.11 ± 0.02) × 10 ^8,b^	(3.34 ± 0.29) × 10 ^8,a^	(3.32 ± 0.12) × 10 ^8,a^	(3.45 ± 0.23) × 10 ^8,a^
15	2-Hexenoic acid, ethyl ester	ND	ND	ND	(1.21 ± 0.21) × 10 ^7^
16	Heptanoic acid, ethyl ester	ND	(1.69 ± 0.11) × 10 ^7,a^	(1.55 ± 0.08) × 10 ^7,ab^	(1.40 ± 0.08) × 10 ^7,b^
17	Octanoic acid, methyl ester	ND	(3.07 ± 0.11) × 10 ^6,b^	(3.34 ± 0.06) × 10 ^6,a^	(3.14 ± 0.06) × 10 ^6,b^
18	Octanoic acid, ethyl ester	(3.73 ± 0.21) × 10 ^8,c^	(6.10 ± 0.57) × 10 ^8,a^	(5.16 ± 0.25) × 10 ^8,ab^	(4.50 ± 0.31) × 10 ^8,bc^
19	Nonanoic acid, ethyl ester	ND	(4.45 ± 0.23) × 10 ^6,b^	(1.43 ± 0.12) × 10 ^7,a^	(3.35 ± 0.12) × 10 ^6,b^
20	Ethyl 9-decenoate	(1.30 ± 0.08) × 10 ^8^	ND	ND	ND
21	Decanoic acid, ethyl ester	(1.80 ± 0.06) × 10 ^8,d^	(2.82 ± 0.02) × 10 ^8,a^	(2.41 ± 0.03) × 10 ^8,b^	(2.04 ± 0.12) × 10 ^8,c^
22	Dodecanoic acid, ethyl ester	(1.90 ± 0.14) × 10 ^7,a^	(9.40 ± 0.22) × 10 ^6,c^	(1.26 ± 0.06) × 10 ^7,b^	(1.22 ± 0.06) × 10 ^7,b^
23	2,2,4-Trimethyl-1,3-pentanediol diisobutyrate	ND	ND	ND	(1.71 ± 0.04) × 10 ^7^
	Total	1.21 × 10 ^9^	2.52 × 10 ^9^	2.24 × 10 ^9^	1.96 × 10 ^9^
	**Aldehydes and ketones**				
24	Acetaldehyde	(1.75 ± 0.04) × 10 ^8,c^	(1.71 ± 0.09) × 10 ^9,a^	(4.26 ± 0.38) × 10 ^8,b^	(4.73 ± 0.44) × 10 ^8,b^
25	5-Hydroxymethylfurfural	(4.58 ± 0.14) × 10 ^7,b^	(3.27 ± 0.25) × 10 ^8,a^	(5.31 ± 0.12) × 10 ^7,b^	(6.19 ± 0.53) × 10 ^7,b^
	**Alkanes**				
26	Ethane, 1,1-diethoxy-	ND	(1.63 ± 0.12) × 10 ^9^	ND	ND
27	Pentane, 1-(1-ethoxyethoxy)-	ND	(1.20 ± 0.07) × 10 ^7^	ND	ND
28	Tridecane	ND	(1.62 ± 0.03) × 10 ^6,a^	(1.53 ± 0.003) × 10 ^6,b^	ND
29	Tetradecane	(2.19 ± 0.05) × 10 ^6,b^	(5.04 ± 0.27) × 10 ^6,a^	(5.12 ± 0.38) × 10 ^6,a^	(2.60 ± 0.07) × 10 ^6,b^
30	Pentadecane	(3.84 ± 0.20) × 10 ^6,b^	(1.00 ± 0.12) × 10 ^7,a^	(6.17 ± 0.48) × 10 ^6,b^	(4.25 ± 0.07) × 10 ^6,b^
31	Hexadecane	(3.76 ± 0.15) × 10 ^6,c^	(9.24 ± 0.80) × 10 ^6,a^	(6.76 ± 0.20) × 10 ^6,b^	(4.37 ± 0.05) × 10 ^6,c^
32	Heptadecane	ND	(3.58 ± 0.23) × 10 ^6^	ND	ND
	Total	9.79 × 10 ^6^	1.67 × 10 ^9^	1.96 × 10 ^7^	1.12 × 10 ^7^
	**Other**				
33	Naphthalene, 2-methyl-	(2.28 ± 0.11) × 10 ^6,a^	(2.17 ± 0.14) × 10 ^6,a^	(2.48 ± 0.19) × 10 ^6,a^	(2.53 ± 0.01) × 10 ^6,a^

Data represent means of three replicates ± standard deviation. ND refers to not detected. Values followed by different letters in each row indicate significant differences (*p* < 0.05). Control represents without the addition of mulberry leaves; 1, 2 and 3% (*w*/*v*, g/mL dry weight) represent the added amount of mulberry leaf powder.

## Data Availability

The authors declare that all data generated or analyzed during this study are included in this published article.

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
