# Peer review of "Mixed Mulberry Fruit and Mulberry Leaf Fermented Alcoholic Beverages: Assessment of Chemical Composition, Antioxidant Capacity In Vitro and Sensory Evaluation"

_foods, 2022, doi:10.3390/foods11193125_

Round 1

Reviewer 1 Report

Comments and Suggestions for Authors

Title: Alcoholic Fermentation of Mixed....authored by Gao et al., Foods 2022.

2.1 Mulberries

There are different levels of color in mulberries. Authors need to idicate why Humulus variety was chosen. This is a very important answer in relation to the decision of using this crop.

2.3.2 Spectrophotometric ...

ABTS was selected to assess the free radical-scavenging activity. Several authors have demonstrated some weak points of this test. Authors are invited to state very clearly the reasons of using only this test. And what about using another complementary test to improve the quality of results?

2.3.7 Determination of volatile substances

Authors are also invited to indicate why this procedure was selected.

2.4 Sensory evaluation

In view of the importance of this test of the research work, authors should briefly describe the basic components  of the training of the sensory assessment group.

2.5 Statistical Analysis

Why Duncan's correlation test was used? There are other tests.

Other tests

In most or all cases authors do not indicate the reason or reasons to use tests involved in this study (e.g., 2.3.1, 2.3.2....). Authors are invited to enrich your report giving attention to this suggestion.

Conclusions

Most of the comments in this Section are based on qualitative type assessments. Authors need to state some quantitative data in order to give support to several conclusions provided here.

In brief, it is an interesting and useful report.

Author Response

Dear reviewer:

We are very grateful to the Editor and reviewers in recognising our work. We acknowledge all the comments and suggestions made by the Editor and reviewers that are very valuable in improving the quality of our manuscript. We have made all the required corrections properly as suggested by the Editor and reviewers. The corrections and amendments have been indicated in a red in the revised manuscript for easy follow-up.

Reviewer 2 Report

Comments and Suggestions for Authors

My comments refer essentially to sensory analysis

Line 76

Was only one fermentation carried out for each sample? Considering the variability of the fermentations it would have been preferable to conduct at least two tests for each of the selected samples

Line 218
The author should provide more information on the organization sensory analysis sessions.
For example: What kind of test was used? How many replicas were made? Was the assessment performed in individual booths?

Furthermore, the number of subjects should be at least 10 people and the panel should be gender-balanced.

Line 514

With regard to the results of the sensory analysis, the author refers to figure 5. Figure 5, however, illustrates the results of the PCA performed on the data of free amino acids.

Furthermore, no reference is made to any statistical analysis.

The author must correct and complete this section.

Author Response

(The authors gave the same response as above.)

Reviewer 3 Report

Comments and Suggestions for Authors

Dear Authors, I have reviewed the article entitled: Alcoholic Fermentation of Mixed Wine Containing Mulberry Fruit and Leaves: Assessment of Phenolics, DNJ, Antioxidant 3 Capacity In Vitro and Sensory Evaluation.

The article is a complex study that describes several variables in the composition of some fermented drinks from mulberry leaves and fruits. After the analysis, I formulated the following recommendations:

1.      Please revise the title to be more concise and taking into account the fact that wines are the drinks obtained by fermenting grapes. Abbreviations should also not appear in the title.

2.      The abstract should be a summary of the methods used in the research. Please review in this regard and name all the methods used. Also, the main objective of the research must be clearly described.

3.      Line 26. When you say that various functional substances are changed, name what they are and what their function is. If you listed how many volatile compounds you determined, say how many amino acids...etc.

4.      Replace the term "wine" with the fermented beverages throughout the text. Or with Fruit Wine at least.

5.      In the introduction you named the bioactive compounds from the leaves, but the polysaccharides are not anti-obesity or anti-atherosclerosis. Please review and indicate precisely for each class of substances what the health benefits are and specify the bibliographic indication.

6.      Be careful that fermented drinks contain alcohol. This is not a health benefit. Please make a clarification in this regard.

7.      Please add the appropriate bibliographic references in the Introduction. For example, number 6 does not contain all the information presented by you in the sequence to which it is attached.

8.      Also, the all Introduction section needs to be improved.

9.      Lines 67-73 should be moved to the Results section. In the materials section we are talking strictly about the raw material.

10.  Line 94: the term oenological parameters is used to characterize fermented drinks from grapes, please reformulate.

11.  Line 96, please replace bibliographic reference 14 with one representative of the methods used.

12.  Line 222 and 223 should not be written "Aroma" and "Palate" in capital letters.

13.  In table 1, the values for sulfur dioxide must be integer values.

14.  The volatile acidity is expressed in grams of acetic acid per liter? Please define.

Author Response

(The authors gave the same response as above.)

Reviewer 4 Report

Comments and Suggestions for Authors

Authors present the effect of addition of mulberry leaves during alcoholic  fermentation  to mulberry wine and studied possible functional properties and sensory characteristics. Manuscript is interesting, novel, well written and presented in general. Below are some queries to authors and some suggestions for improvement.

1. Is there any regulation concenring addition of leaves to such wine? why 3% is considered high? have authors tried higher amounts of leaves addition?

2.Authors performed 9 day fermentation, why not more or less days, please explain or add reference/s. Did author check the presence of mycotoxins in the leaves used? 

3. for TPC results were expressed as g gallic acid /L mulberry wine. In the literature mg or g GAE/L is usually used, please explain the use of specific expression.

4. mulberry suspension was maintained at 50 °C, how authors retained stable temperature? Also 50 degrees seems too high for extraction, above 40 degrees compounds can be deteriorated, please explain.

5. i suggest authors to add a paragraph with all standard compounds used, company of purchase for all assays used and HPLC. For example which anthocyanins where used as stds? Also how quantification of phenolics was performed by HPLC? type of detector and model are missing, please add for all methods for phenolics and DNJ and GABA.

6. {lease describe in brief Hitachi L-8900 automatic amino 201 acid analyzer (Hitachi Ltd., Japan) analysis, e.g. validation, calibration curves, daily QC work. also about The volatile substances that were analyzed by the Atomax Teklink software control purging sample concentration system, not clear how work was done in this case, give brief explanation. For GCMS analysis did authors use libraries for identificationof compounds?

7.for sensory evaluation at least 8=10 people are needed. in this case 7 were tested samples, please explain. Does the pamel accepted acidity icrease mentioned at p. 243? at p. 322 it is written that no significant difference in acidity was observed.

8. for ABTS and FRAP i suggest to use only one decimal digit unless authors explain the need for two.

9. fig 1 and 2 for the high concentrations, in black colour, bottom error bars are not visible and significant differences or not cannot be chacked.

10. sensory evaluation results are vague, not clear why specific aromas were observed. mulberry leaves were dried in a convection drying oven...this is not an explanation. pleae revise.

General comments: authors should present chromatograms for HPLC and GCMS analysis indicative of each application, even as supplement materials. How quantification was done is not clear. Last but not least it is not mentioned if sensory evaluators accepted [psitively attributes of leaves added to wine. 2% addition said to be nutricious, why, please rephrase or explain.

Author Response

(The authors gave the same response as above.)

Reviewer 5 Report

Comments and Suggestions for Authors

Even though the manuscript “Alcoholic Fermentation of Mixed Wine Containing Mulberry Fruit and Leaves: Assessment of Phenolics, DNJ, Antioxidant Capacity In Vitro and Sensory Evaluation” could be of interest to the Foods readers, I believe that major changes should be made to the manuscript. In general terms, the discussion should be improved. Some specific recommendations are enlisted bellow:

In the title and in the abstract, the authors should indicate what DNJ stands for.

Through all document, authors should use extracted instead of dissolved, when talking about the bioactive compounds extracted from the leaf powder.

In the introduction, how rich in nutrients are the mulberry leaves? I believe that they are rich in bioactive compounds, but not in nutrients.

In methodology, I was expecting that authors were going to characterize the mulberry leave powder, and they should explain why they used 1, 2 and 3% values. Also, it is not clear what does this percentage stands for: 1 g leave/99 g fruit?, or por 100 mL juice? I really believe that dry fruit and leaves should be characterized in order to be able to discuss their results in terms of these samples.

I recommend that authors should determine the total sugar content during fermentation. This is relevant, since sugars may interact with the Folin reactive overestimating the total phenolic content. I believe that if authors consider the effect of sugar on total phenolic content, their results will better correlate with those of individual phenolic compounds determined by HPLC.

Determination of total anthocyanin content (Equation 1) is confusing, it is not clear why authors determined anthocyanins by different methods. In reference 11, the authors didn’t determine monomeric and polymeric anthocyanins. It is not clear why if in reference 11, authors used the pH differential method, in this study they used a different method. Tartrate and flavonoids determination (line 116) is confusing, how specific for flavonoids and tartrate are these methods? Why caffeic acid was used as tartrate standard.

Section 2.3.3 should be revised, it is confusing, why two different HPLC conditions are reported. Does the extraction procedure is specific for flavonols compounds?

In section 2.3.4 line 173 for the determination of DNJ, was the precipitate collected, or it was the supernatant?

Section 2.3.6 should be revised, it appears that amino acids were determined only in the dried fruits, however in results the wine samples are reported.

In both HPLC and GC-MS, authors should indicate which compounds were used as standards.

Section 2.4 should be described with more detail.

In section 2.5 PCA and heatmapping should be described with more detail.

In lines 263-265, authors are indicating that total sugar content was lower in leaf fermented wines, however, the opposite is observed in table 1.

In all tables, the authors should indicate what the capital and non-capital letters mean in each column

In table 1 (and in methodology), it is not clear why the monomeric, polymeric, and total anthocyanins are determined as AU

In section 3.2, the authors should evaluate the effect of sugars on the determination of phenolic compounds through the Folin method. It has been reported that the presence of sugar in the samples overestimates the total phenolic content. It is recommended to pass the sample through a C18-SPE cartridge to remove sugar prior to phenolic content determination. See for instance

Muñoz-Bernal, 2020

DOI: https://doi.org/10.21548/41-1-3778

This section should be discussed in more detail, explaining which will be the mechanism involved in the modification on the bioactive content due to different leave addition as well as fermentation time. For instance, in line 330, authors should discuss why the addition of mulberry leaf powder reduces the levels of cyanidin-3-O-rutinoside and glucoside.

Section 3.3.1 should be revised; authors are discussing different anthocyanins than those reported on table 3.

All tables should be revised. why do all samples have the same initial values in all tables? I was expecting that the addition of the leaf extracts will modify the initial concentrations

Section 3.3.2 should be revised, the discussion does not agree with the reported values in table 3

The GABA section discussion should be improved, how relevant are the reported values?

Paragraph in lines 427-429 is confusing.

Figure 3 should be better described and discussed, its confusing.

I believe that Figures 5A and 5B were wrongly selected, these figures should be of the sensory analysis, not PCA.

Author Response

(The authors gave the same response as above.)

Round 2

Reviewer 3 Report

Comments and Suggestions for Authors

The authors made the required corrections. The work may be published. 

Author Response

Dear Reviewer:

    We are very grateful to the Editor and reviewers in recognising our work. We acknowledge all the comments and suggestions made by the Editor and reviewers that are very valuable in improving the quality of our manuscript. We checked and corrected the manuscript again. The corrections and amendments have been indicated in blue in the revised manuscript for easy follow-up.

Reviewer 4 Report

Comments and Suggestions for Authors

Authors replied to my comments adequately and followed my suggestions to improve their manuscript,  I am satisfied with their response. Hence, paper can now be published in its current form. 

Author Response

Dear Reviewer:
    We are very grateful to the reviewers in recognising our work. 

Kindest regards,

Dr. Jie Yang (yangjie0737@163.com)

Associate professor

Jiangsu Key Laboratory of Marine Bioresources and Environment/Jiangsu Key Laboratory of Marine Biotechnology

Jiangsu Ocean University

Lianyungang, China

Reviewer 5 Report

Comments and Suggestions for Authors

The quality of the manuscript was improved, however, there are some corrections that should be made to the manuscript. For instance, in line 380, the authors should provide a reference that supports the idea that fermentation accelerates the decomposition of anthocyanins

The recommendation given in revision 1 “All tables should be revised. why do all samples have the same initial values in all tables? I was expecting that the addition of the leaf extracts will modify the initial concentrations” should be addressed, I understand that control samples should be the same, however, at time 0, it is not possible that all samples have the exact same concentration than that of the control. I was expecting that the initial (time 0) values increased as the addition of leave extract increased from 1 to 3%.

Author Response

Dear Reviewer:

   We are very grateful to the Editor and reviewers in recognising our work. We acknowledge all the comments and suggestions made by the Editor and reviewers that are very valuable in improving the quality of our manuscript. We have made all the required corrections properly as suggested by the Editor and reviewers. The corrections and amendments have been indicated in blue in the revised manuscript for easy follow-up.
